# Conditional Latent Space Molecular Scaffold Optimization for Accelerated Molecular Design

**Onur Boyar**                                                  *boyar.onur.nagoyaml@gmail.com*
*Nagoya University*

**Hiroyuki Hanada**                                  *hanada.hiroyuki.i9@f.mail.nagoya-u.ac.jp*
*Nagoya University*

**Ichiro Takeuchi**                                  *takeuchi.ichiro.n6@f.mail.nagoya-u.ac.jp*
*Nagoya University*
*RIKEN*

**Reviewed on OpenReview:** *https://openreview.net/forum?id=KhxVc9RBJv*

## Abstract

The rapid discovery of new chemical compounds is essential for advancing global health and developing treatments. While generative models show promise in creating novel molecules, challenges remain in ensuring the real-world applicability of these molecules and finding such molecules efficiently. To address this challenge, we introduce Conditional Latent Space Molecular Scaffold Optimization (CLaSMO), which integrates a Conditional Variational Autoencoder (CVAE) with Latent Space Bayesian Optimization (LSBO) to strategically modify molecules while preserving similarity to the original input, effectively framing the task as constrained optimization. Our LSBO setting improves the sample-efficiency of the molecular optimization, and our modification approach helps us to obtain molecules with higher chances of real-world applicability. CLaSMO explores substructures of molecules in a sample-efficient manner by performing BO in the latent space of a CVAE conditioned on the atomic environment of the molecule to be optimized. Our extensive evaluations across diverse optimization tasks—including rediscovery, docking score, and multi-property optimization—show that CLaSMO efficiently enhances target properties, delivers remarkable sample-efficiency crucial for resource-limited applications while considering molecular similarity constraints, achieves state of the art performance, and maintains practical synthetic accessibility. We also provide an open-source web application[1] that enables chemical experts to apply CLaSMO in a Human-in-the-Loop setting.

## 1 Introduction

The accelerated discovery of chemical compounds represents a crucial challenge with the potential to revolutionize global health, offering new ways to combat diseases and viruses. The ability to efficiently discover and develop new chemical compounds could lead to groundbreaking treatments and therapies, addressing some of the most pressing health issues of our time. As the importance of this field grows, so too does the research focused on finding effective solutions. Over the past few years, artificial intelligence (AI) has emerged as a powerful tool in this endeavor. The combination of increased computational power and advancements in generative modeling has brought us closer than ever to achieving significant breakthroughs in accelerated discovery.

Generative models offer various approaches to exploring and creating new chemical compounds. A common strategy involves training a generative model on a comprehensive database of chemical compounds. Once

---

[1]The source code of this work is available at https://github.com/onurboyar/CLASMO-TMLR.

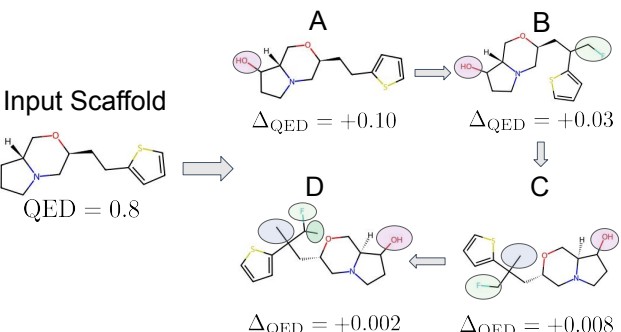

Figure 1: An example of the CLaSMO framework updating a scaffold for Quantitative Estimation of Drug-likeness (QED) optimization. CLaSMO identifies optimal regions in the CVAE latent space and selects bonding points on the scaffold. Chemical features from these points are used to guide the generation of substructures, which are then integrated into the scaffold (A, B, C, D) through small, targeted modifications to improve molecular properties.

trained, the model can generate entirely new compounds (Gómez-Bombarelli et al., 2018; Tripp et al., 2020; Griffiths & Hernández-Lobato, 2020; Boyar & Takeuchi, 2024; Grosnit et al., 2021; Boyar et al., 2024). This approach opens the door to discovering novel molecules that are unlike any currently known, potentially revealing a vast, untapped universe of chemical possibilities. However, while the generation of these novel molecules is exciting, their practical applicability is often constrained. The challenges lie in synthesizing these novel molecules and in the limited understanding of their properties, which makes it difficult for domain experts to assess their viability (Lim et al., 2020).

To address these challenges, another strategy for designing new molecules focuses on modifying/editing existing compounds using generative models (Bradshaw et al., 2019; Lim et al., 2020), reinforcement learning (Gottipati et al., 2020), genetic algorithms (Jensen, 2019), or from the domain experts themselves by trial and error (Bemis & Murcko, 1996; Schreiber, 2000; Welsch et al., 2010). These approaches are more likely to produce synthesizable molecules because the base molecule is known to exist and, therefore, can be more tractable for real-world applications. However, even with this strategy, a significant obstacle remains: sample-efficiency, i.e., how efficiently a method finds a promising molecular modification with a limited number of molecular property evaluations. Evaluating the properties of a chemical compound is a time-consuming and costly process, and many existing methods in the literature require numerous trials, making them less practical for accelerated discovery. Consequently, there is a critical need for a methodology that can generate target chemical compounds in a more sample-efficient manner.

In this study, we introduce the Conditional Latent Space Molecular Scaffold Optimization (CLaSMO) method, a framework designed to address two critical challenges in chemical compound discovery: real-world applicability and sample-efficiency. CLaSMO combines a Conditional Variational Autoencoder (CVAE) (Higgins et al., 2016) with Latent Space Bayesian Optimization (LSBO) (Gómez-Bombarelli et al., 2018) to strategically modify input molecules and optimize their chemical properties.

In drug discovery, a common strategy for generating synthesizable molecules is to work with molecular scaffolds—key substructures that serve as the foundation for chemical modifications and drug design (Bemis & Murcko, 1996). Building on this approach, our framework integrates small substructures into these scaffolds to improve key molecular properties. For modifications to be both effective and synthesizable, it's crucial that the generative model understands how new substructures bond with the scaffold and enhance its properties. To achieve this, we condition substructure generation on the scaffold's chemical features using a CVAE. Our novel data preparation and training strategy enables the CVAE to generate substructures that align with specific atomic environments. We further optimize this process using LSBO, which efficiently explores both the latent space of the CVAE and the scaffold's chemical features. This allows CLaSMO to effectively select regions for modification and generate substructures that are chemically meaningful. By tracking molecular similarity between the initial scaffold and the optimized molecule, CLaSMO is designed to operate in low-

budget scenarios and costly settings, such as wet-lab experiments, where high sample-efficiency is crucial. It ensures that modifications are both realistic and applicable to real-world contexts. Ultimately, this approach accelerates the discovery of novel, synthesizable compounds with improved properties. Figure 1 illustrates an example of molecular optimization with CLaSMO.

We evaluate our approach on a diverse suite of 20 molecular optimization tasks that span a wide range of objectives, including rediscovery of known compounds, multi-property optimization, and drug-likeness enhancement. These tasks present different challenges, enabling a comprehensive assessment of our model's performance across varied molecular design scenarios. To further test robustness, we incorporate experimental settings with varying levels of similarity constraints between input and optimized molecules, reflecting both flexible and constrained optimization regimes. In addition to property optimization, we also assess the synthetic and retrosynthetic accessibility of the generated molecules. Our results show that CLaSMO consistently improves target properties in a sample-efficient manner, demonstrating strong generalizability and effectiveness across these diverse tasks.

1. We propose CLaSMO, a pioneering CVAE and LSBO-based molecule modification algorithm for molecular design. We use a novel data preparation strategy that enables CVAE to learn how substructures bond with target molecules, providing tailored generations.

2. We show that CLaSMO improves target properties with sample-efficiency while keeping the optimized molecules structurally similar to the input scaffolds, increasing the likelihood of identifying synthesizable compounds with desirable properties.

3. We open source a web-application, https://clasmo.streamlit.app/, that enables interactive optimization of input molecules via CLaSMO, which allows chemical experts to decide the region to modify in input molecule, enabling Human-in-the-Loop optimization settings.

## 2 Related Works

Molecular design strategies can broadly be divided into two categories: from-scratch-generation of molecules and modification-based approaches. Both categories have made significant strides in recent years, yet they also face unique challenges, particularly regarding real-world applicability and sample-efficiency.

From-scratch-generation approaches focus on creating entirely new molecules by conducting a search to optimize the target property. A seminal work by Gómez-Bombarelli et al. (2018) introduced latent space optimization-based methodology, using a VAE to generate novel compounds by navigating the latent space of molecular representations. LSBO builds on this by efficiently reducing the number of expensive black-box evaluations required for molecular optimization, enabling the discovery of compounds with desirable properties in a continuous latent space. Since then, numerous studies have further refined and expanded the LSBO framework, focusing on method development and practical applications (Grosnit et al., 2021; Tripp et al., 2020; Maus et al., 2022; Griffiths & Hernández-Lobato, 2020; Boyar & Takeuchi, 2024; Boyar et al., 2024). However, like many other generation-from-scratch methods, such methodologies struggle with real-world applicability—i.e., the difficulty of synthesizing the generated molecules in real-world settings (Lim et al., 2020). Other generation methods, such as genetic algorithms (Jensen, 2019), Grammar VAE (Kusner et al., 2017), and Junction Tree (JT) VAE (Jin et al., 2018), aim to improve the chemical validity of generated structures. Recent advancements like GP-MOLFORMER (Ross et al., 2024) utilizes large language model-like approaches for molecular design, but the real-world applicability challenge remains a major limitation across these methodologies.

Modification-based approaches, on the other hand, focus on adding substructures to existing molecules or scaffolds, often leading to more synthesizable and interpretable designs. Methods like Scaffold-GGM (Lim et al., 2020) employ a graph generative model to modify molecular scaffolds, thus improving properties with a higher chance of obtaining synthesizable molecules. Weller & Rohs (2024) introduces DrugHIVE, a deep hierarchical variational autoencoder that leverages scaffold modification to generate novel molecular compounds. A concurrent study by Lee et al. (2025) proposes a diffusion model based approach for both from-scratch and scaffold modification-based generation problems. There are many other scaffold-based

optimization methodologies (Li et al., 2019; Langevin et al., 2020), not limited to generative modeling (Schreiber, 2000; Welsch et al., 2010; Miao et al., 2011). Techniques like Bradshaw et al. (2019)'s model ensure chemical validity in molecular modifications, and Gottipati et al. (2020)'s PGFS model uses reinforcement learning to guide additions to the base molecule. These approaches aim to avoid the low real-world viability problem faced by generation-from-scratch methods, as they build upon known molecular scaffolds, however, they lack advanced optimization methodologies that take sample-efficiency into account.

CLaSMO combines the strengths of both categories, leveraging LSBO in the latent space of a CVAE for improved sample-efficiency while focusing on scaffold-based modifications. Unlike current LSBO-based molecular design methodologies in the literature, CLaSMO does not generate molecules from scratch but instead optimizes molecular properties by adding substructures to existing scaffolds. This approach mitigates the real-world applicability problem, and increases the chance of obtaining molecules that are both effective and practical for real-world synthesis. To evaluate the sample-efficiency of CLaSMO and benchmark it against established methodologies, we follow the sample-efficiency benchmark introduced by Gao et al. (2022). This benchmark provides a suite of challenging molecular property optimization tasks, using a variety of oracle functions, and compares the performance of numerous molecular design methods from the literature.

A key group of methods included in this benchmark are genetic algorithm-based approaches, which remain competitive and widely applicable to scaffold optimization. Notable examples include Graph-GA (Jensen, 2019), which uses graph-based genetic algorithms in combination with a generative model; Smiles-GA (Brown et al., 2019), which operates on SMILES string representations (Weininger, 1988); and STONED (Nigam et al., 2021), which leverages SELFIES (Krenn et al., 2020), a robust alternative to SMILES for molecular encoding. These methods have demonstrated strong performance and robustness, and benchmarking against them is highly recommended, as emphasized in Tripp & Hernández-Lobato (2023).

Beyond genetic algorithms, other scaffold-optimization-capable methodologies have also gained attention. For example, MARS Xie et al. (2021) employs a Markov Chain Monte Carlo (MCMC) approach with an annealing scheme and adaptive proposals to iteratively edit molecular graph fragments, guided by a graph neural network. It has demonstrated strong performance across various tasks, particularly in multi-objective optimization scenarios where properties such as bioactivity, drug-likeness, and synthesizability must be jointly optimized. Additionally, reinforcement learning techniques such as MolDQN (Zhou et al., 2019b) have proven effective for property optimization tasks. Recent advances in GFlowNets (Bengio et al., 2021) have also opened new avenues in molecular design. For instance, SynFlowNet (Cretu et al., 2024), a GFlowNet-based framework, supports scaffold optimization and has demonstrated competitive performance.

We adopt and modify the experimental setting proposed by Gao et al. (2022) in Section 5.2 to better align with the scope of our work—namely, molecular optimization in low-budget scenarios, where oracle evaluations are limited and sample-efficiency is essential. Within this setting, we demonstrate that CLaSMO consistently outperforms baseline methodologies across a diverse set of tasks. By combining a targeted scaffold-modification strategy with efficient search, CLaSMO offers a balanced and effective solution to molecular optimization under constrained evaluation budgets.

To assess the synthesizability of the generated molecules, we report both the Synthetic Accessibility score (SA score) Ertl & Schuffenhauer (2009) and the Retrosynthetic Accessibility score (RA score)(Thakkar et al., 2021). The SA score is obtained via estimation of synthetic feasibility based on the occurrence of molecular fragments in publicly available databases, serving as a fast and interpretable proxy for synthetic feasibility. However, it may underestimate difficulty for molecules with rare or unconventional motifs, or overestimate it for simple but impractical structures. To address such limitations, we complement this metric with the RA score, which reflects the predicted success of retrosynthetic route planning based on a classifier trained on reaction network outcomes. By incorporating both metrics, we obtain a more comprehensive assessment: SA scores reflect the structural simplicity and prevalence of a molecule in known chemical space, while RA scores provide insight into practical synthesis routes using available chemistry knowledge. This dual evaluation is particularly critical in low-data settings, where optimization strategies must not only find high-performing molecules, but also prioritize those amenable to real-world synthesis.

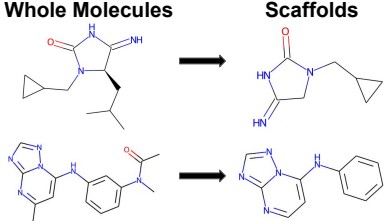

Figure 2: Examples of scaffold extraction from whole molecules. In both the upper and lower rows, side chains are removed, leaving the core structure of the molecule. These resulting scaffolds act as starting points for novel chemical design.

## 3 Preliminaries and Problem Setup

In this section, we provide preliminary knowledge on CVAEs, LSBO, and scaffolds. We then discuss the challenges of property optimization and scaffold modifications.

### 3.1 Conditional Variational Autoencoders (CVAEs)

A VAE (Kingma & Welling, 2014) consists of an encoder $f_\phi^{enc} : \mathcal{X} \to \mathcal{Z}$ and a decoder $f_\theta^{dec} : \mathcal{Z} \to \mathcal{X}$, where $\mathcal{X}$ represents the input space and $\mathcal{Z}$ the latent space. CVAEs extend the framework of VAEs by incorporating additional condition vector $\boldsymbol{c}$ into the latent variable model, facilitating the controlled generation of new instances. In the CVAE architecture, the encoder $q_\phi(\boldsymbol{z}|\boldsymbol{x},\boldsymbol{c})$ maps an input $\boldsymbol{x}$ and a condition $\boldsymbol{c}$ to a latent representation $\boldsymbol{z}$. Simultaneously, the decoder $p_\theta(\boldsymbol{x}|\boldsymbol{z},\boldsymbol{c})$ reconstructs $\boldsymbol{x}$ using both $\boldsymbol{z}$ and $\boldsymbol{c}$. The training of CVAEs is formulated as the minimization of the conditional variational lower bound:

$$\mathbf{J}(\theta,\phi;\boldsymbol{x},\boldsymbol{c}) = -\mathbb{E}_{q_\phi(\boldsymbol{z}|\boldsymbol{x},\boldsymbol{c})}\left[\log p_\theta(\boldsymbol{x}|\boldsymbol{z},\boldsymbol{c})\right] + \mathrm{KL}\left(q_\phi(\boldsymbol{z}|\boldsymbol{x},\boldsymbol{c}) \parallel p(\boldsymbol{z})\right), \tag{1}$$

where KL denotes the Kullback-Leibler divergence. In this model, the prior distribution $p(\boldsymbol{z})$ over the latent variables is typically assumed to be a standard normal distribution, $\mathcal{N}(0,I)$. This assumption simplifies the learning process by standardizing the latent space, ensuring that the encoder learns a distribution that closely aligns with a prior distribution, thus enhancing the generative capability of the decoder conditioned on specific contexts.

### 3.2 Latent Space Bayesian Optimization (LSBO)

In BO, we start with numerous *unlabeled* instances $\{\boldsymbol{x}_i\}_{i\in[\mathcal{U}]}$ and a smaller set of *labeled* instances $(\boldsymbol{x}_i, y_i)_{i\in[\mathcal{L}]}$, where an input $\boldsymbol{x}_i \in \mathcal{X}$ represents a chemical compound, and a label $y_i \in \mathcal{Y} \subseteq \mathcal{R}$ indicates its properties such as docking scores. BO seeks to optimize a costly black-box function $f^{BB} : \mathcal{X} \to \mathcal{Y}$, which corresponds to obtaining the physical properties of chemical compounds through experiments or time-consuming simulations in the context of molecular design problems. The goal is to maximize $f^{BB}$ with minimal evaluations, using typically a Gaussian Process (GP) surrogate, trained using the labeled instances $\mathcal{L}$, to predict the function over $\mathcal{X}$. BO uses the surrogate to select an input $\boldsymbol{x}$ that may yield values surpassing the current maximum $\max_{i\in\mathcal{L}} y_i$. However, building a GP surrogate in high-dimensional spaces like chemical compounds is challenging. LSBO tackles this by employing a VAE/CVAE trained on the unlabelled instances $\mathcal{U}$ to reduce dimensionality, encoding instance in $\mathcal{X}$ to lower dimensional latent space $\mathcal{Z}$. This simplifies surrogate modeling and optimization because $\mathcal{Z}$ has lower-dimension than $\mathcal{X}$. During LSBO iterations, the acquisition function applied to the GP's predictions selects new points in $\mathcal{Z}$ to evaluate. The chosen latent variable $\boldsymbol{z}_{i'}$ is decoded into a new input $\boldsymbol{x}_{i'} = f_\theta^{dec}(\boldsymbol{z}_{i'})$. This new instance is evaluated by $f^{BB}$, and the results update $\mathcal{L}$ and refine the GP model. This cycle repeats until optimal results are achieved or resources are exhausted. In contexts like molecular design, LSBO aims to discover chemical compounds with optimal properties by efficiently navigating the reduced latent space.

### 3.3 Scaffolds

Scaffolds (Bemis & Murcko, 1996) are the stable core structures within molecules that serve as the framework for chemical modifications in drug design. Scaffolds retain the essential biological activity of the molecule. They play a crucial role in molecular design by providing a foundation for chemical modifications aimed at optimizing properties like QED. Researchers often use scaffolds to systematically explore chemical variations (Schreiber, 2000; Welsch et al., 2010), which can lead to the discovery of new compounds with improved properties.

In this study, we follow the scaffold extraction method from Bemis & Murcko (1996), where non-essential components like side chains are removed, leaving the core structure. This extracted scaffold serves as a starting point for further modifications, allowing efficient exploration of chemical space. By focusing on scaffolds, molecular design becomes more streamlined, increasing the likelihood of identifying novel, synthesizable compounds with the desired biological activity. Examples of whole molecule and scaffold pairs are provided in Fig. 2.

### 3.4 Problem Definition

We denote the scaffold, which is the base of the modification, as $S$, and the modified molecule as $S'$. Our goal is to efficiently find the modification that maximizes the molecular property P which is evaluated by $f^{BB}(S')$, while keeping the difference between $S$ and $S'$ small. Directly iterating over all possible $S'$ to find the best modification that maximizes $f^{BB}(S')$ is impractical, as it involves high complexity and costly evaluations of $f^{BB}$.

The primary challenge in optimizing molecular scaffolds lies in i) determining the optimal bonding point on the base scaffold $S$, and ii) selecting the appropriate substructures added to the bonding point to ensure meaningful improvements in the desired property $\mathcal{P}$. A molecular scaffold $S$, composed of several atoms $p_1, p_2, \ldots, p_k$, may have atoms with the remaining capacity to form additional chemical bonds. These atoms serve as potential candidates for bonding with newly generated substructures. Therefore, the task involves not only selecting the right substructure but also identifying the most suitable bonding point $p_i$ to optimize scaffold properties. This adds complexity, as the need for precise modifications must be balanced with the challenges of high-dimensional search spaces and evaluation costs. Consequently, a more efficient approach is required to explore scaffold modifications effectively while minimizing the number of evaluations.

To address this, the problem can be reframed as an optimization task in a reduced latent space $\mathcal{Z}$, obtained through a CVAE. In this space, each point $z \in \mathcal{Z}$ corresponds to a potential substructure that can be integrated into the scaffold. By encoding the molecular substructures into this lower-dimensional space $\mathcal{Z} \in \mathbb{R}^d$, the search becomes more tractable. The objective is to find the optimal latent representation $z^*$ conditioned on the optimal bonding point $p_i^*$ that, together, maximize the desired property $\mathcal{P}$ when the generated substructure $s' \leftarrow f^{dec}(z)$ is added to the scaffold. Let us denote this modification as $S' \leftarrow g(S \oplus s', p_i)$, where the function $g()$ adds substructure $s$ to the scaffold $S$ at $p_i$. The optimization problem is then formulated as:

$$z^*, p_i^* = \arg\max_{z \in \mathcal{Z}, p_i \in B(\mathbf{S})} f^{BB}(S') = \arg\max_{z \in \mathcal{Z}, p_i \in B(\mathbf{S})} f^{BB}(g(S \oplus f^{dec}(z), p_i)), \tag{2}$$

where $B(S)$ is the set of possible bonding points on the scaffold $S$. In Section 4.2, we demonstrate that atomic features at $p_i$ are used as condition vectors to generate new substructures, enabling targeted substructure generation for atom $p_i$.

#### 3.4.1 Controlling Molecular Similarity

Current modification-based methods often fail to account for how changes impact molecular similarity between the original scaffold $S$ and the updated scaffold $S'$, or any structure in general. Adding substructures typically increases molecular weight, which can hinder real-world applicability, especially when exceeding 500 Daltons, as indicated by Lipinski's Rule of Five (Lipinski et al., 2001). Higher molecular weight compounds are more difficult to synthesize, making them less suitable for molecular design. Additionally, it is sometimes necessary to ensure that modifications result in only minor adjustments to avoid drastic changes.

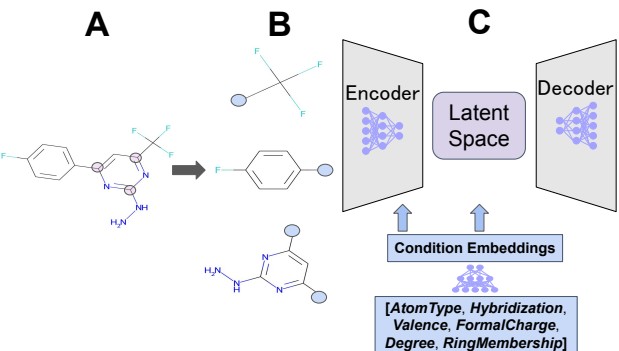

Figure 3: Illustration of the BRICS algorithm and its integration with the CVAE. The molecule in **A** is decomposed into substructures using the BRICS algorithm (**B**), with specific breaking points highlighted. Atomic environment features are extracted from these points to provide critical information about the bonding environment. Embeddings of these features are used as condition vectors, which are concatenated both at the encoder input along with the substructures and at the latent space of the CVAE (**C**), guiding the generation of substructures that are compatible with the scaffold.

Thus, a key challenge is to guide the optimization process by considering molecular similarity, ensuring that the modified molecules remain structurally close to the original scaffold. Such a framework can increase the likelihood of obtaining synthesizable compounds by limiting divergence from known molecules. In Section 5, we show that our method effectively solves the optimization problem in Eq. (2) while ensuring molecular similarity between $S$ and $S'$.

# 4    Proposed Method

Our proposed CLaSMO framework comprises two key components: the CVAE and the LSBO algorithm. However, an essential first step in our approach is the data preparation required to train the CVAE. In this section, we will begin by outlining the data preparation process, followed by an explanation of the CVAE and the CLaSMO methodology.

## 4.1    Data Preparation

Our proposed method requires a uniquely tailored dataset because no existing dataset in the literature fully meets the specific needs of our approach. To create this dataset, we developed a BRICS (*Breaking Retrosynthetically Interesting Chemical Substructures*) (Degen et al., 2008) based approach. BRICS is an algorithm designed to decompose organic molecules into smaller, synthetically feasible substructures by identifying breaking points within a molecule's structure based on chemical retrosynthetic rules. These breaking points, also known as division points, are the connections between subgraphs within the molecule that BRICS identifies. The algorithm systematically breaks down a molecule $M$ into $k$ substructures, Fig. 3 demonstrates this procedure. BRICS is particularly well-suited for this task because it ensures that the generated substructures are synthetically feasible and chemically valid, making it ideal for scaffold-based molecular optimization. Unlike other decomposition methods, BRICS explicitly adheres to retrosynthetic rules, ensuring compatibility with real-world chemical synthesis.

The division points found by BRICS are of particular importance because they serve as both the points where the molecule is split into substructures and the potential bonding sites where these substructures can be reattached. Therefore, they provide us a valuable information about these substructures in terms of what kind of bonds they can form. This dual role makes the division points a crucial piece of information for our generative model. By preserving the properties and features of atoms at these division points, we capture the chemical context around the atom that dictates how substructures can bond with other parts of a molecule. The atomic features we consider are *atom type, hybridization, valence, formal charge, degree, and ring membership.* Detailed explanations of these features are provided in the Appendix A.1.

These features, when used as condition vectors in our CVAE, are essential for guiding the generation of substructures that can successfully bond with the scaffold, as they provide a detailed understanding of the chemical environment at each division point, such as their atom types and if the selected atom has the capacity for forming additional bonds. Therefore, our dataset for CVAE training includes the substructures obtained via the BRICS algorithm and six different atomic features of the atom at their breaking points. These substructures are represented using a string-based representation method referred to as SELFIES Krenn et al. (2020). The illustration of the BRICS algorithm, atomic feature extraction points, and their integration with CVAE is provided in Fig. 3.

## 4.2 Substructure CVAE

Our CVAE consists of an encoder, parameterized by $f_\phi^{\text{enc}}$, and a decoder, parameterized by $f_\theta^{\text{dec}}$. The encoder takes the substructure $s$ and the associated condition vector $c$, and maps this input into a latent space, producing a latent representation $z$. The decoder then takes a point $z$ from this latent space, along with the condition vector $c$, and generates a substructure $s'$ that is conditioned on the given atomic properties. The loss function of the proposed CVAE is defined as:

$$\mathbf{J}(\phi, \theta; \mathbf{s}, c) = \|s - \mathbb{E}_{q_\phi(z|s,c)}[p_\theta(s|z,c)]\|^2 + \beta \cdot D_{\text{KL}}(q_\phi(z|s,c)\|p(z)), \tag{3}$$

where $q_\phi(z|s, c)$ is the encoder, $p_\theta(s|z, c)$ is the decoder, and $\beta$ balances the reconstruction and KL divergence terms (Higgins et al., 2016). The condition vectors allow the CVAE to learn how different substructures interact with specific atomic environments, building a deeper understanding of their bonding behavior. Figure 3B demonstrates the input substructures that CVAE uses in its training, and Fig. 3C visualizes the CVAE architecture along with the condition vectors.

After training, if during the generation phase, we provide the CVAE with a condition vector that specifies the atomic environment of the target scaffold, the decoder, using a latent vector along with the condition vector, generates substructures that are tailored to bond effectively with the scaffold. This conditioning mechanism ensures that the generated substructures are not random but specifically designed to fit the scaffold's atomic environment, improving the efficiency of scaffold modifications and the likelihood of successful bonding. In the proposed CLaSMO method, we jointly optimize the bonding position $p_i$ and the latent vector $z$, conditioned on the atomic properties $c$ of the bonding site in order to optimize both the bonding position and the substructure added to the position.

### 4.2.1 Condition Vector Embeddings

In Section 4.1, we detailed our data preparation process and the extraction of six atomic environmental features, resulting in a 6-dimensional condition vector. As we will explain in Section 5.1, the relatively simple nature of the substructures used in CVAE training allows for a much lower latent dimension than the actual conditional vector dimension of 6. To align with this simplicity and ensure efficient representation, we employed an Autoencoder model to generate $d$-dimensional embeddings of the atomic features. These embeddings preserve the key characteristics within the atomic environments and learn the relationship among distinct atomic features while reducing dimensionality, which helps lower computational complexity and improves the model's ability to generalize. After its training, the encoder of the Autoencoder, $f_{\text{CondEmd}}^{\text{enc}}$, is used to provide the final condition vector $c$ to be inputted to CVAE by encoding the six-dimensional atomic environmental feature vector. This process is visualized in Fig. 3C.

## 4.3 CLaSMO

In CLaSMO, upon training of the CVAE, we perform a targeted search in the latent space $\mathcal{Z} \in \mathbb{R}^d$ of the CVAE, where decodings from each point $z$ represent a potential substructure to be added to the scaffold. The challenge is to modify the scaffold $S$ by selecting a substructure and integrating it at an appropriate bonding point $p_i$, optimizing the desired molecular property $\mathcal{P}$. Therefore, the search space $\Omega$ we consider in our optimization tasks is defined as the Cartesian product of the latent space and the set of possible bonding points on the scaffold:

$$\Omega = \mathbb{R}^d \times \{p_1, p_2, \ldots, p_n\}.$$

Within this search space, the goal is to modify the scaffold $\boldsymbol{S}$ to maximize the BB function $f^{\mathrm{BB}}(\boldsymbol{S'})$.

However, although our main goal is optimizing the target property, we also aim to keep the similarity between $\boldsymbol{S}$ and $\boldsymbol{S'}$ at a certain level. Therefore, in CLaSMO, we apply a similarity constraint using the *Dice Similarity* (Dice, 1945) metric to compare the input scaffold $\boldsymbol{S}$ with the modified molecule $\boldsymbol{S'}$ before evaluating the BB function. In order to measure the similarity between $\boldsymbol{S}$ and $\boldsymbol{S'}$, we use Morgan Fingerprints (Morgan, 1965; Rogers & Hahn, 2010). The Morgan fingerprint of a molecule is represented as a binary vector, where each bit indicates the presence or absence of a specific substructure within the molecule, making them effective and popular for comparing molecular similarities. Using these, the similarity between $\boldsymbol{S}$ and $\boldsymbol{S'}$ is measured by computing the Dice Similarity of their Morgan fingerprint vectors[2], defined as:

$$\mathrm{DICE}(\boldsymbol{S}, \boldsymbol{S'}) = \frac{2|\boldsymbol{M_S} \cap \boldsymbol{M'_S}|}{|\boldsymbol{M_S}| + |\boldsymbol{M'_S}|}$$

where $\boldsymbol{M_S}$ and $\boldsymbol{M'_S}$ are the Morgan fingerprint vectors of $\boldsymbol{S}$ and $\boldsymbol{S'}$, respectively. Dice Similarity, ranging from 0 (no overlap) to 1 (identical), helps us track structural changes during optimization.

The final optimization objective, incorporating both the search for optimal substructures and the similarity constraint, is given by:

$$\boldsymbol{z^*}, p_i^* = \arg \max_{(\boldsymbol{z}, p_i) \in \Omega} f^{\mathrm{BB}}(\boldsymbol{S'}) \tag{4}$$

$$\text{subject to:} \quad \mathrm{DICE}(\boldsymbol{S}, \boldsymbol{S'}) \geq \tau,$$

where $\boldsymbol{S'} \leftarrow g(\boldsymbol{S} \oplus f^{\mathrm{dec}}(\boldsymbol{s'}|\boldsymbol{z^*}, \boldsymbol{c}))$. By this approach, CLaSMO efficiently navigates the latent space, optimizing molecular properties while allowing modifications only if $\mathrm{DICE}(\boldsymbol{S}, \boldsymbol{S'}) \geq \tau$, effectively controlling the degree of divergence from the input scaffold.

For the joint optimization of substructures and bonding positions, we consider a GP model: $(\boldsymbol{z}, p) \mapsto y'_\Delta$, where $\boldsymbol{z}$ is the latent vector representing a substructure, $p$ is the bonding position of the modification, and $y'_\Delta := y - y'$ represents the improvement in the property, with $y$ and $y'$ indicating the properties of molecules $S$ and $S'$, respectively. To prepare the training dataset $\mathcal{D}$ for the GP, we conducted a random sampling of substructures from random latent vectors $\boldsymbol{z}$, each paired with randomly selected bonding regions $p$ on the scaffold $\boldsymbol{S}$, and evaluated the $y'_\Delta$ obtained from these additions (e.g., in the case of QED optimization, QED score differences between $\boldsymbol{S}$ and $\boldsymbol{S'}$ are calculated). These triplets of $(\boldsymbol{z}, p, y'_\Delta)$ are used in GP training. This setup allows the GP model to learn the relationship between the latent space representations, atoms in the input scaffold, and the resulting property changes, guiding the optimization process toward regions of the latent space that are more likely to yield beneficial modifications to the scaffold.

Among many possible choices, we employ the Upper Confidence Bound (UCB) acquisition function to guide the optimization process. To ensure LSBO samples from regions that meet the similarity constraint, we introduced a penalization mechanism. Specifically, we assign a negative improvement value $y'_\Delta$ when the condition $\mathrm{DICE}(\boldsymbol{S}, \boldsymbol{S'}) \geq \tau$ is not satisfied after sampling from $f^{\mathrm{dec}}([\boldsymbol{z^*}, \boldsymbol{c}])$. Additionally, we also apply a penalty when a substructure cannot bond with the target molecule. We outline our approach in Algorithm 1, and provide details about the selection of hyperparameters within our framework in the Appendix A.3.

### 4.3.1 Kernel Design of CLaSMO

The problem we try to solve requires simultaneous optimization over continuous latent vectors and discrete bonding points. To handle this complexity, we use a GP model with a covariance function $k$ that accommodates the mixed input space of continuous and discrete variables. We define separate kernels for these inputs:

$$k_{\mathrm{cont}}(\boldsymbol{z}, \boldsymbol{z'}) = \exp\left(-\frac{1}{2\ell_1^2}\|\boldsymbol{z} - \boldsymbol{z'}\|^2\right), \quad k_{\mathrm{cat}}(p_i, p_j) = \exp\left(-\frac{\delta_{p_i, p_j}}{\ell_2}\right),$$

---

[2]Bajusz et al. (2015) discusses that the dice similarity is one of the best metrics to assess similarity between molecules using Morgan fingerprints.

where $k_{\text{cont}}(\boldsymbol{z}, \boldsymbol{z}')$ is an RBF kernel, and $k_{\text{cat}}(p_i, p_j)$ measures the similarity between atoms in the molecule that are ready for additional bond via Kronecker delta function, measuring equality of atom positions, $\ell_1$ and $\ell_2$ are the lengthscale parameters for the continuous and categorical components, respectively. The combined kernel used in CLaSMO is then expressed as:

$$k_{\text{CLaSMO}} = k((\boldsymbol{z}, p_i), (\boldsymbol{z}', p_j)) = k_{\text{cont}}(\boldsymbol{z}, \boldsymbol{z}') \cdot k_{\text{cat}}(p_i, p_j).$$

---

**Algorithm 1** CLaSMO
---
1: **Input:** GP training data $\mathcal{D}$, Trained CVAE, Trained Autoencoder encoder $f_{\text{CondEmd}}^{\text{enc}}$ for condition embeddings, Input molecules $\boldsymbol{M}$, Optimization budget per molecule $K$, Similarity threshold $\tau$, Penalization terms $\lambda_1, \lambda_2$
2: Fit GP using $\mathcal{D}$
3: **for** $i = 1$ to $\boldsymbol{M}$ **do**
4:     Pick molecule $\boldsymbol{m}_i$, obtain its scaffold $\boldsymbol{S}_i \leftarrow \text{Scaffold}(\boldsymbol{m}_i)$,
5:     Evaluate target property value $y \leftarrow f^{\text{BB}}(\boldsymbol{m}_i)$
6:     **for** $j = 1$ to $K$ **do**
7:         Identify available atoms in the scaffolds for bonding, $p_j \in B(\boldsymbol{S})^2$
8:         Find $\boldsymbol{z}^*, p_i^* = \arg\max_{(\boldsymbol{z}_i, p_i) \in \Omega} f^{\text{BB}}(S')$
9:         Create condition vector $\boldsymbol{c}^*$ for atom $p_i^*$ in molecule $\boldsymbol{S}_i$
10:        Obtain condition embeddings $\boldsymbol{c} \leftarrow f_{\text{CondEmd}}^{\text{enc}}(\boldsymbol{c}^*)$
11:        Generate substructure $\boldsymbol{s}^* \leftarrow f^{\text{dec}}([\boldsymbol{z}^*, \boldsymbol{c}])$
12:        Add substructure $\boldsymbol{s}^*$ to molecule $\boldsymbol{S}_i$ at region $p_i^*$ to obtain $\boldsymbol{S}_i'$
13:        **if** $\boldsymbol{S}_i \neq \boldsymbol{S}_i'$ **then**
14:            **if** $\text{DICE}(\boldsymbol{S}_i, \boldsymbol{S}_i') > \tau$ **then**
15:                Evaluate new property: $y' = f^{\text{BB}}(\boldsymbol{S}_i')$
16:                Compute improvement $y_\Delta' = (y - y')$
17:                **if** $y_\Delta' > 0$ **then**
18:                     Update $\boldsymbol{S}_i \leftarrow \boldsymbol{S}_i'$
19:                **end if**
20:            **else**
21:                Set $y_\Delta'$ to penalization term $\lambda_1$
22:            **end if**
23:        **else**
24:            Set $y_\Delta'$ to penalization term $\lambda_2$
25:        **end if**
26:        Update $\mathcal{D} \leftarrow \mathcal{D} \cup \{[\boldsymbol{z}^*, p^*], y_\Delta'\}$
27:        Update GP with $\mathcal{D}$
28:     **end for**
29: **end for**

---

# 5 Experiments

In this section, we first describe the training procedure for our CVAE model (Section 5.1). We start from our dataset selection procedure, and introduce the details of our model architecture. We then evaluate the performance of CLaSMO across 20 molecular property optimization tasks Gao et al. (2022), which span a range of objectives including rediscovery of known compounds, similarity-based optimization, multi-property optimization, and improvements in drug-likeness.

To assess robustness across different optimization regimes, each task is repeated under three levels of molecular similarity constraints between the input and optimized molecules, defined by Dice similarity thresholds

---

[2] This identification is performed by finding atoms within the molecule that have additional capacity to form bonds. For example, a carbon atom can form up to four bonds, and if it has only three bonds within the molecule, it will be identified as available for bonding.

of 0.50, 0.25, and 0. This results in a total of 60 optimization settings. Higher thresholds enforce stricter structural similarity, enabling controlled, local edits, while lower thresholds allow for broader exploration of chemical space. We also evaluate the synthetic and retrosynthetic accessibilities of the generated molecules to assess their feasibility from a chemical synthesis and retrosynthetic planning perspective.

Section 5.2 presents the results of these optimization experiments and the corresponding synthetic and retrosynthetic accessibility analysis. In Section 5.3, we provide an ablation study to examine the impact of conditional generation. Finally, Section 5.4 discusses the extension of CLaSMO to a human-in-the-loop optimization setting.

## 5.1 CVAE Training

In this section, we start from the dataset selection procedure and then provide model training details.

### 5.1.1 Dataset Selection

Our motivation for using a BRICS-curated dataset is twofold. First, BRICS decomposition enables interpretable fragmentation of molecules by producing chemically meaningful substructures with explicit connection points. These points can be directly associated with atomic features such as atom type, degree, and hybridization, which allows us to condition the generative model on the chemical context at each attachment site. Second, since our overall objective is to develop a generative model that performs small, targeted modifications to existing molecules, it is essential to train on substructures that reflect realistic, low-molecular-weight chemical fragments. This encourages the model to propose minimal, context-aware edits to molecular scaffolds during inference.

To this end, we evaluated two widely used open-source molecular datasets: QM9 (Ruddigkeit et al., 2012; Ramakrishnan et al., 2014) and ZINC250K (Gómez-Bombarelli et al., 2018). We applied BRICS decomposition to both datasets and analyzed the resulting fragment distributions with a focus on their suitability in our setting. As shown in Figure 4, BRICS fragments derived from QM9 are compact and chemically consistent, with all fragments under 105 Da and most well below 100 Da. In contrast, fragments from ZINC250K exhibit greater chemical complexity and structural variability, with a substantial portion exceeding 100 Da in molecular weight. Representative examples of fragments from both datasets are also presented in Figure 4.

Based on this analysis, we selected QM9 as our primary dataset for training the CVAE model. QM9 contains approximately 130,000 small organic molecules and, when processed via BRICS, yields 18,706 unique substructure pairs along with their atomic environment features. The relatively small size and chemical simplicity of these fragments align well with our modeling objective of generating small, synthetically accessible modifications to existing scaffolds. Additionally, the compact nature of QM9 fragments eliminates the need for further filtering, making the dataset highly consistent and easy to work with. This makes QM9 particularly suitable for our use case, where the goal is to preserve high similarity to the input scaffold while introducing meaningful yet minimal structural improvements[4]

### 5.1.2 Model Training

Among the 18,706 instances, 80% were used for model training, with the remaining data allocated for testing and validation. We represented the molecules using SELFIES (Krenn et al., 2020), a string-based molecular representation, in the form of one-hot encoding matrices. Our CVAE architecture consists of three fully connected layers in the encoder and three GRU layers in the decoder. Given the simplicity of the dataset, we determined that a 2-dimensional latent space was sufficient to achieve over 90% reconstruction accuracy on the test set, where training is conducted using an early stopping condition along with a learning rate scheduler. This lower-dimensional space also enhances the performance of BO, which typically excels in smaller-dimensional spaces. See Appendix A.3 for further details on the selection of hyperparameters in CLaSMO and in training the CVAE model.

---

[4]In Appendix A.2, we explore two possible extensions: one where the CVAE is trained exclusively on BRICS fragments derived from the ZINC250K dataset, and another where it is trained on a mixture of QM9 fragments and size-compatible fragments filtered from ZINC250K.

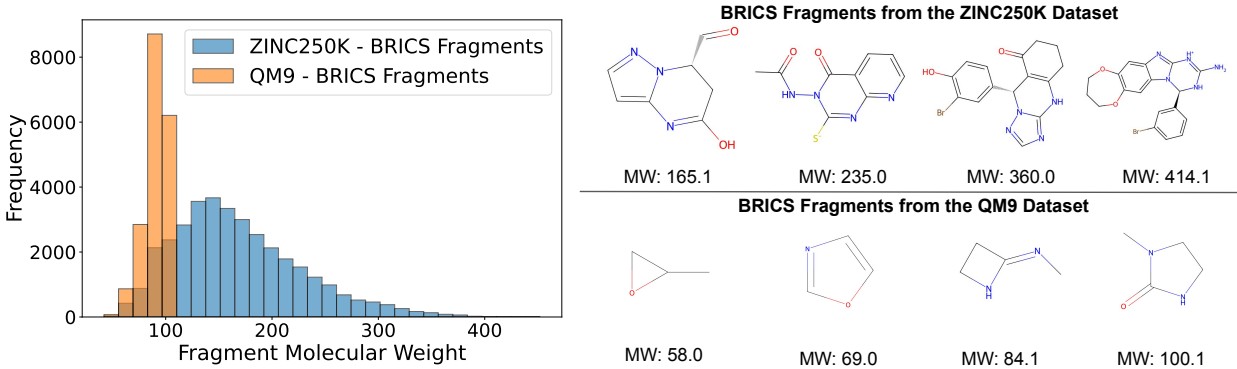

Figure 4: Distribution of BRICS fragment molecular weights from the QM9 and ZINC250K datasets, along with representative examples from each. BRICS fragments from the QM9 dataset generally have lower molecular weights, reflecting its focus on smaller molecules, while the ZINC250K dataset contributes a broader range of fragment sizes.

Additionally, to maintain this simplicity, as explained in Section 4.2.1, we generated a 2-dimensional embedding for the condition vectors using an Autoencoder model, which is trained with fully connected layers in both the encoder and decoder, achieved 93% reconstruction accuracy for the condition vectors. Prior to CVAE training, we obtained the condition vector embeddings via the Autoencoder model, and used them during CVAE training. As a result, the CVAE was trained with both a 2-dimensional latent space for the substructures and 2-dimensional condition vectors, effectively balancing simplicity and optimization performance. Further discussion on this design choice is provided in Appendix A.4.

## 5.2 Molecular Optimization Experiments

To comprehensively evaluate the sample-efficiency of our methodology, we applied CLaSMO to the sample-efficiency benchmark proposed by Gao et al. (2022), where we use 20 diverse molecular optimization tasks. These tasks span various objectives, including a broad range of pharmaceutically-relevant oracle functions, and challenging multi-property optimization (MPO) scenarios (Brown et al., 2019). Instead of limiting the evaluation to narrowly defined tasks, this broad selection enables a more comprehensive assessment of each method's generalizability and adaptability across chemically and functionally diverse scenarios.

Here, sample-efficiency refers to the efficiency with respect to the number of evaluations of the black-box function for the target optimization or editing task. For instance, when evaluating molecules through wet-lab experiments, it is often the case that no more than about 100 evaluations can be conducted. Our proposed method, CLaSMO, is specifically designed to achieve such sample-efficient molecular editing. On the other hand, in the numerical experiments presented in this paper, we selected problems where the evaluation of the target black-box functions can be performed at relatively low computational cost, allowing for repeated and comprehensive comparisons between the proposed method and existing methods. Hereafter, we refer to these functions as oracle functions. Experiments were conducted with a fixed budget of 100 oracle evaluations per seed across 10 seeds, ensuring consistent comparisons. The choice of 100 oracle iterations (in contrast to the 10,000 iterations used in Gao et al. (2022)) reflects our focus on developing a methodology tailored for resource-constrained scenarios, which are more representative of real-world applications.

CLaSMO was benchmarked against six widely used baseline methods—Smiles-GA (Brown et al., 2019), Graph-GA (Jensen, 2019), Stoned (Nigam et al., 2021), MolDQN (Zhou et al., 2019a), MARS (Xie et al., 2021), and SynFlowNet (Cretu et al., 2024)—using the same set of 100 scaffolds for all methods as starting molecules. To assess optimization performance across scaffolds of varying sizes, we randomly sampled input scaffolds from both the QM9 and ZINC250K datasets. Appendix A.5 provides detailed statistics on their molecular weight and property value distributions, along with visualizations of all 100 scaffolds used in the evaluation. As explained below, different similarity constraints were applied, and open-source implemen-

tations of all baseline methods were used to evaluate their ability to perform molecular optimization while preserving the given scaffold. We note that some of these baseline methods were not designed for sample-efficient settings, and therefore may not perform well under the problem settings aimed at sample-efficient molecular editing, which is the focus of this study.[5]

### 5.2.1 Oracle Definitions and Grouping

To facilitate clearer comparisons and highlight performance patterns, we group the 20 oracles used in our experimental setting into four task-oriented categories. The groups and their associated oracle functions are as follows:

- **Group 1: Similarity and Rediscovery Tasks** This group comprises oracles that either assess molecular similarity to a known drug or target the rediscovery of a known molecule. The oracles in this group include: *albuterol_similarity, celecoxib_rediscovery, mestranol_similarity, thiothixene_rediscovery, troglitazone_rediscovery.*

- **Group 2: Multi-Property Optimization (MPO) Tasks** These involve optimizing multiple molecular properties simultaneously. The oracles in this group include: *osimertinib_mpo, perindopril_mpo, ranolazine_mpo, sitagliptin_mpo, zaleplon_mpo.*

- **Group 3: Structural Modification Tasks** This group encompasses tasks that evaluate a model's ability to generate structurally diverse molecules while preserving specific chemical features. The *hopping* tasks challenge models to modify the molecule while aiming to maintain or enhance biological activity despite structural changes. In contrast, the *isomers* tasks focus on generating different structural arrangements (isomers) of a molecule that share the same molecular formula but differ in connectivity or spatial orientation. Unlike similarity-based tasks that prioritize closeness to a reference molecule, these tasks emphasize structural innovation and diversity, assessing a model's capacity to explore novel chemical spaces while retaining desired properties. The oracles in this group include: *deco_hop, scaffold_hop, isomers_c7h8n2o2, isomers_c9h10n2o2pf2cl.*

- **Group 4: Activity and Benchmark Tasks** This group includes oracles that predict biological activity or are part of broader benchmarking suites for drug-likeness and diversity. The oracles in this group include: *drd2, gsk3b, median1, median2, qed, valsartan_smarts.*

Further details and descriptions of these oracles are provided in Appendix A.7.[6]

### 5.2.2 Property Optimization Results

In this section, we discuss the performance of CLaSMO and benchmark methodologies under three similarity-constraint regimes. In the high constraint setting ($\tau = 0.50$), the optimized molecule must have a Dice similarity of at least 0.50 to the input scaffold. In the moderate constraint setting ($\tau = 0.25$), a similarity of at least 0.25 is required, allowing for more flexibility in modifying the scaffold. In these settings, even if the target property can be substantially improved, the result will not be accepted if the similarity falls below the specified threshold. Lastly, in the no constraint setting ($\tau = 0$), there are no restrictions on similarity, and any substructure additions to the scaffold are allowed[7].

**Optimization with High Constraints (Similarity Threshold $\tau = 0.50$)** Figure 5 shows the performance of the methods when the similarity constraint is very strict.

- In the **Similarity and Rediscovery Tasks**, CLaSMO is consistently the top performer, providing the top performance from 5 tasks out of 5.

---

[5]Among the competing approaches, Graph-GA, Smiles-GA, and Stoned lack a mechanism to ensure preservation of the given scaffold. We implemented a hyperparameter optimization for these genetic algorithm-based approaches to find the best balance for exploration of the search space and scaffold-preserving capabilities. Please refer to Appendix A.6 for further details on experimental setting.

[6]See also the Therapeutics Data Commons for more detail: `https://tdcommons.ai/functions/oracles/`.

[7]See Appendix A.8 for a further discussion of the rationale behind the chosen similarity constraint thresholds.

- Within the **MPO Tasks**, CLaSMO and MARS methodologies are the two top performers. MARS, a method that is specifically designed for MPO settings, outperforms CLaSMO in two out of 5 MPO objectives, and provides a similar performance in *osimertinib_mpo* task.

- For the **Structural Tasks**, CLaSMO gives the best results in *deco_hop, scaffold_hop* and *isomers_c7h8n2o2* while MARS achieves a similar result with CLaSMO in the remaining *isomers_c9h10n2o2pf2cl* task.

- In the **Activity and Benchmark Tasks**, CLaSMO records the highest values for *drd2, median1* and *qed*, while losing to MARS in *gsk3b* and *median2*. None of the methodologies provided an improvement in *valsartan_smarts* task.

Overall, under high constraints, CLaSMO proves to be the best approach, providing the highest Top-10 average scores in 13 out of 20 property optimization tasks.

**Optimization with Moderate Constraints (Similarity Threshold $\tau = 0.25$)** Figure 6 summarizes the results under a moderate similarity constraint, which allows for more chemical modification while still preserving significant scaffold features.

- In the **Similarity and Rediscovery Tasks**, CLaSMO maintains its lead, where it achieves the highest Top-10 average scores in 4 out of 5 tasks, losing to MARS only in *thiothixene_rediscovery*.

- Among the **MPO Tasks**, MARS is the top performer, where it provides the highest Top-10 average scores in 3 out of 5 tasks. CLaSMO outperforms MARS in only *zaleplon_mpo* task, and performs similarly with MARS in *osimertinib_mpo* task.

- In the **Structural Tasks**, CLaSMO gives the best results in all of the 4 tasks in this group.

- In the **Activity and Benchmark Tasks**, CLaSMO records the highest values for *median1, median2* and *qed*, while losing to MARS in *drd2* and *gsk3b*. None of the methodologies provided an improvement in *valsartan_smarts* task.

Thus, even with moderate flexibility, CLaSMO is able to harness the allowed modifications to deliver the best overall results, providing the best results in 13 optimization tasks out of 20.

**Optimization with No Constraints (Similarity Threshold $\tau = 0$)** When no similarity constraints are imposed (see Fig. 7), methods only need to keep the scaffold, and all additions to the scaffold are allowed.

- **Similarity and Rediscovery Tasks**, CLaSMO maintains its lead, where it achieves the highest Top-10 average scores in 4 out of 5 tasks, losing to MARS only in *thiothixene_rediscovery*.

- In the **MPO Tasks**, MARS gets the best results in 3 out of 5 objectives, and CLaSMO finds the best results in *zaleplon_mpo* task and performs similar as MARS in *osimertinin_mpo* task.

- In the **Structural Tasks**, CLaSMO gives the best results in all of the 4 tasks in this group.

- In the **Activity and Benchmark Tasks**, CLaSMO records the highest values for *median1* and *qed*, while losing to MARS in *drd2, gsk3b* and *median2*. None of the methodologies provided an improvement in *valsartan_smarts* task.

In summary, even without similarity constraints, CLaSMO effectively explores the chemical space to optimize molecular properties with high sample-efficiency. In this setting, MARS and Graph-GA methodologies provides more competitive results compared to constrained optimization settings. However, CLaSMO still achieves the best performance across 11 out of 20 optimization tasks[8]

---

[8]In CLaSMO, the ratio of generated substructures that successfully bond with the given scaffold serves as an important metric for assessing optimization performance. Appendix A.9 provides a detailed analysis of bonding success rates throughout the optimization process.

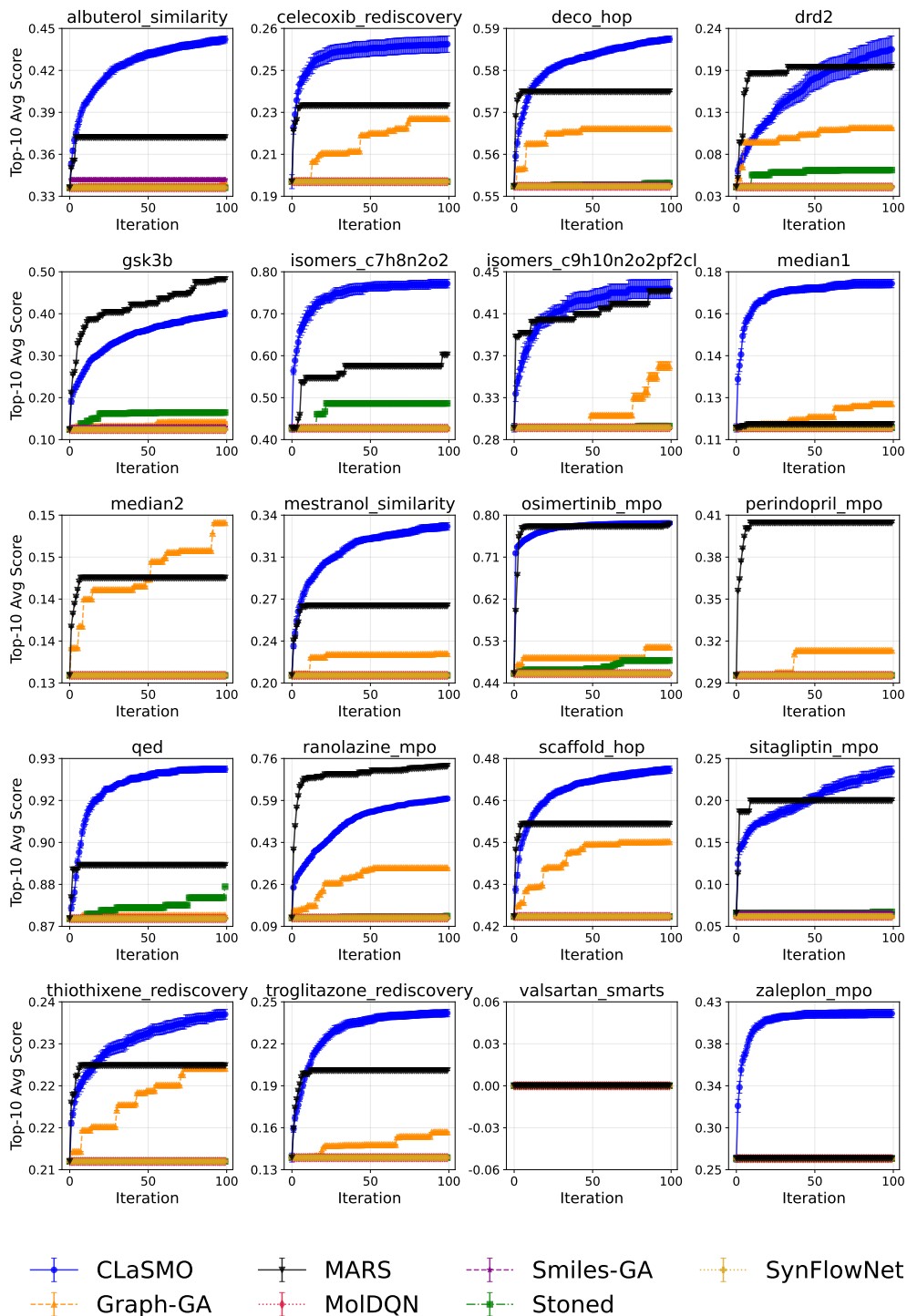

Figure 5: Comparison of Top-10 average values obtained at each iteration for CLaSMO, MARS, Smiles-GA, SynFlowNet, Graph-GA, MolDQN, and Stoned using constraint optimization setting, where Dice Similarity threshold $\tau = 0.50$. CLaSMO identifies molecules with a higher Top-10 average score after 100 iterations in most of the optimization tasks. Among benchmark methodologies, MARS is competitive, followed by Graph-GA and Stoned.

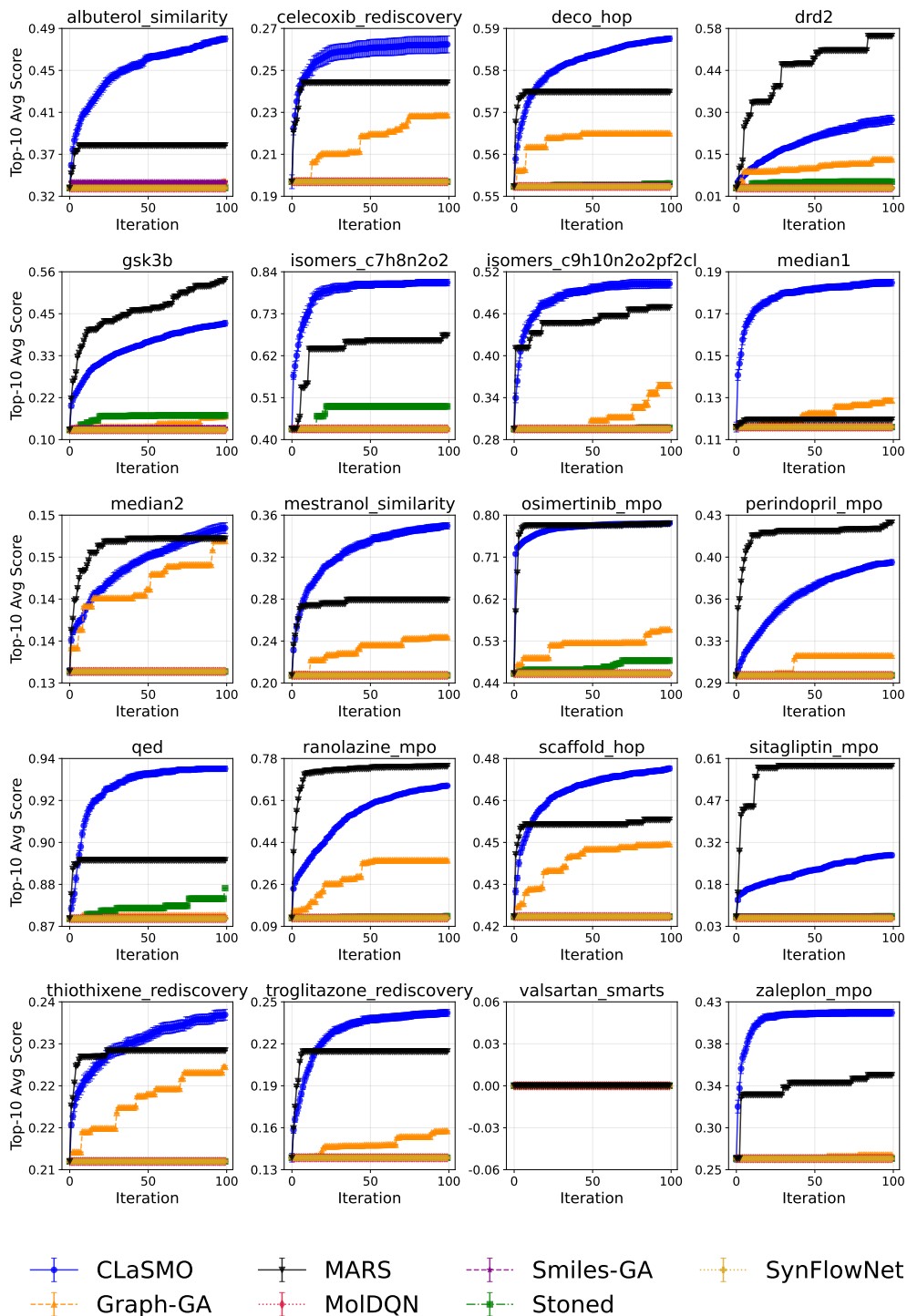

Figure 6: Comparison of Top-10 average values obtained at each iteration for CLaSMO, MARS, Smiles-GA, SynFlowNet, Graph-GA, MolDQN, and Stoned using constraint optimization setting, where Dice Similarity threshold $\tau = 0.25$. CLaSMO identifies molecules with a higher Top-10 average score after 100 iterations in most of the optimization tasks. Among benchmark methodologies, MARS is competitive, followed by Graph-GA and Stoned.

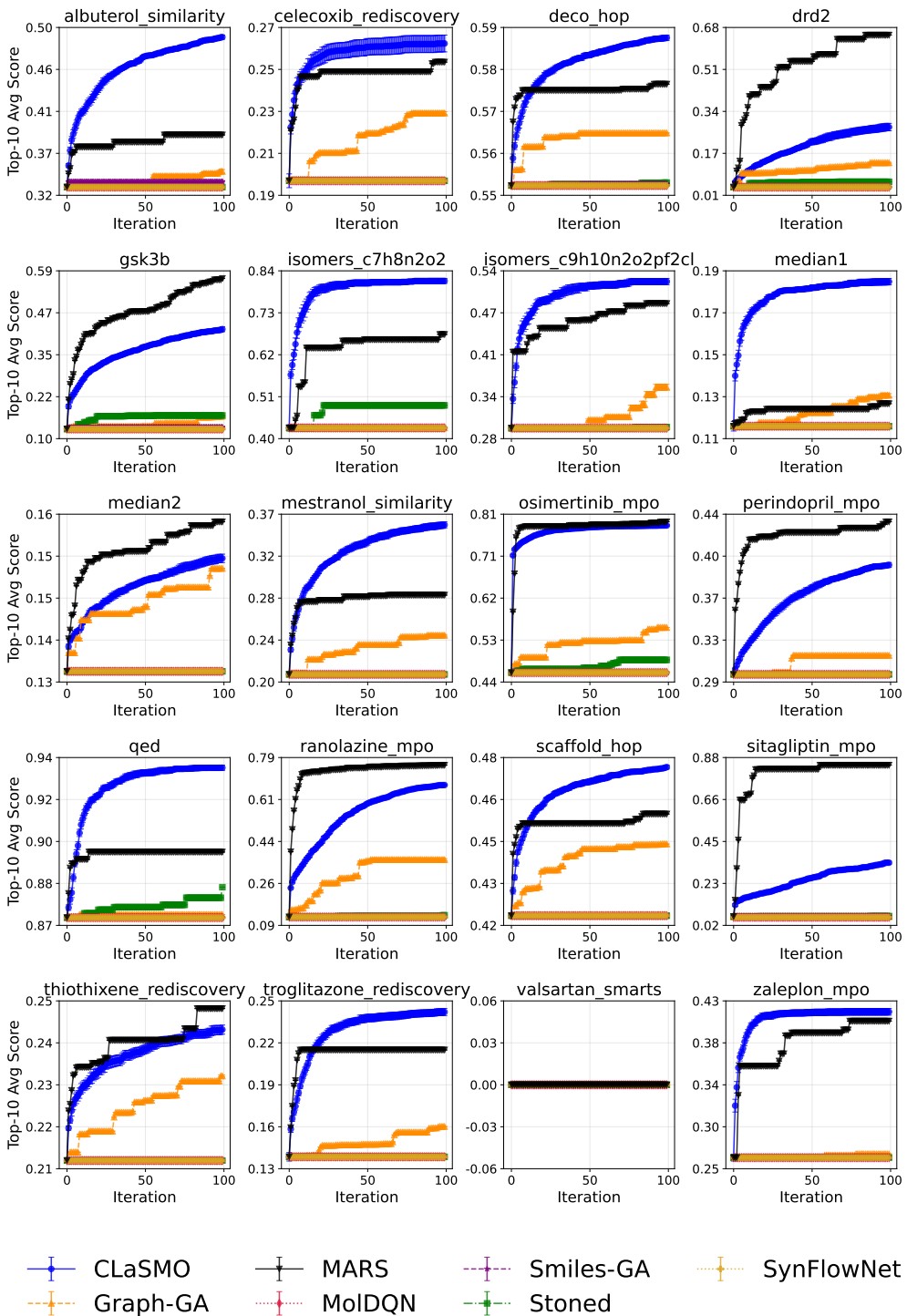

Figure 7: Comparison of Top-10 average values obtained at each iteration for CLaSMO, MARS, Smiles-GA, SynFlowNet, Graph-GA, MolDQN, and Stoned, without any constraint threshold. CLaSMO identifies molecules with a higher Top-10 average score after 100 iterations in most of the optimization tasks. Among benchmark methodologies, MARS is highly competitive, followed by Graph-GA and Stoned.

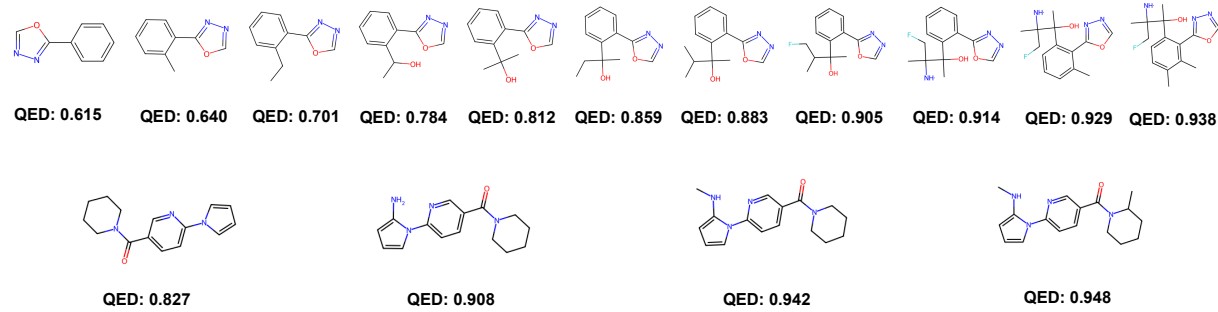

Figure 8: Examples of molecules obtained from different CLaSMO experiments in the QED setting. In both rows, the molecules on the left represent the input scaffolds. Each molecule to the right shows a step of substructure addition along with its resulting QED score. The results demonstrate QED improvements from 0.615 to 0.938 and from 0.827 to 0.948 in these examples.

Across all experimental settings, we observe notable differences in sample-efficiency among the benchmarked methods. MolDQN, a reinforcement learning based approach, and SynFlowNet, a GFlowNet-based approach consistently require substantially more oracle evaluations to achieve comparable optimization levels. These findings are consistent with those reported by (Gao et al., 2022), which highlighted the low property optimization capabilities of such methods[9]. Our results further reinforce that their slower convergence makes them less suitable for resource-constrained sample-efficient optimization tasks, where a low evaluation budget is available. In contrast, genetic algorithm-based methods, especially Graph-GA, remain competitive and often perform well across a variety of tasks. MARS, on the other hand, also shows strong performance, particularly in multi-property optimization scenarios, and achieves solid results overall[10]. However, CLaSMO consistently outperforms all baselines across the majority of tasks and similarity constraint levels. These results highlight CLaSMO's robustness and efficiency across both constrained and unconstrained optimization settings, where sample-efficiency is essential due to limited evaluation budgets.

To illustrate how CLaSMO optimizes input scaffolds, Fig. 8 presents two representative examples from the QED optimization task. In each case, CLaSMO incrementally modifies the input scaffold by adding substructures that are conditionally generated based on the local atomic environment. Each step in the optimization process yields a molecule with an improved QED score, demonstrating the model's ability to progressively refine structure–property relationships. The examples show QED improvements from 0.615 to 0.938 and from 0.827 to 0.948, underscoring CLaSMO's ability to produce chemically valid and property-enhancing modifications through its conditional generation mechanism and LSBO-guided search.

### 5.2.3 Synthetic and Retrosynthetic Accessibility Evaluation

To assess the synthesizability of the optimized molecules, we analyzed both their synthetic accessibility (SA) scores and retrosynthetic accessibility (RA) scores.

The SA score estimates how difficult a compound is to synthesize based on structural complexity and fragment contributions. Lower SA scores indicate molecules that are easier to synthesize, with values typically ranging from 1 (very easy) to 10 (very difficult). Scores above 5 are generally considered indicative of challenging synthesis Anede et al. (2024).

---

[9]SynFlowNet, unlike MolDQN, was not included in the benchmark conducted by Gao et al. (2022). However, other GFlowNet-based methodologies evaluated in that study exhibited lower optimization performance compared to alternative approaches.

[10]Further discussion on the performance comparison on MARS and CLaSMO is provided in Appendix A.10

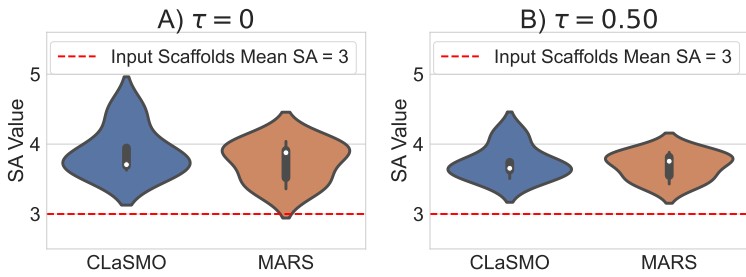

Figure 9: Distributions of average SA scores for molecules optimized by CLaSMO and MARS under similarity constraints $\tau = 0$ (**A**) and $\tau = 0.50$ (**B**). Lower scores correspond to higher synthesizability.

Given that CLaSMO and MARS exhibited the strongest performance in property optimization, we focused our SA and RA analysis on these two methods[11]. We compared the average SA scores obtained under the lowest and highest similarity constraints, $\tau \in \{0, 0.50\}$. Figure 9 presents the distributions of these scores. The horizontal dashed line represents the average SA score of the input scaffolds, which is 3.00.

Both methods generate molecules that are, on average, more complex than the input scaffolds, as the optimized molecules exhibit higher molecular weight and structural complexity. While such increases can pose challenges for synthesis, our results demonstrate that with targeted, small-scale modifications, good synthesizability can still be maintained. At $\tau = 0$, the average SA score is 3.88 for CLaSMO and 3.74 for MARS. At $\tau = 0.50$, these scores improve slightly to 3.73 and 3.68, respectively. Although MARS consistently produces molecules with slightly better synthetic accessibility, CLaSMO maintains competitive synthesizability while achieving stronger performance in property optimization. As indicated by the distribution of SA scores, the generated molecules consistently score below 5, suggesting that their synthesis is likely to be straightforward (Anede et al., 2024).

To complement this structure-based evaluation, we further report the RA score, which offers a model-driven estimate of a compound's synthesizability based on its likelihood of being successfully deconstructed by retrosynthetic planning algorithms. The RA score is computed using a machine learning classifier trained on retrosynthesis outcomes and ranges from 0 to 1, with higher values indicating a higher predicted probability that a valid synthetic route exists. In our evaluation, we report the percentage of molecules that are classified as synthesizable.

As shown in Figure 10, 100% of the input scaffolds are deemed synthesizable. Among the optimized molecules from the experiments with similarity threshold $\tau = 0.5$, 88.0% of those generated by CLaSMO and 90.1% of those from MARS are classified as synthesizable. For the case where there is no similarity threshold, $\tau = 0$, 79.2% of those generated by CLaSMO and 80.6% of those from MARS are classified as synthesizable. These results support the notion that both methods yield molecules that are not only structurally accessible (as evidenced by favorable SA scores) but also recognized by data-driven retrosynthetic models as viable targets for synthesis.

Together, the SA and RA evaluations provide a robust and complementary view of synthesizability. While SA scores reflect fragment-level and structural complexity, RA scores offer a practical assessment from the perspective of route planning. CLaSMO, in particular, achieves a compelling balance, delivering high amount of property improvements across many tasks while maintaining strong performance across both accessibility metrics and similarity constraints. These findings reinforce its potential for real-world molecular design scenarios where synthetic feasibility is essential.

---

[11]There is a strong correlation between the SA score and the similarity to the original molecule. When the similarity to the original molecule is high, the SA score also tends to be favorable, as the original molecule is synthesizable. Conversely, when the similarity is low, the SA score tends to be unfavorable. As a result, molecules generated by methods that involve minimal modifications to the original molecule tend to receive good SA scores. Therefore, we focus on comparing CLaSMO and MARS, both of which achieve high scores in property optimization.

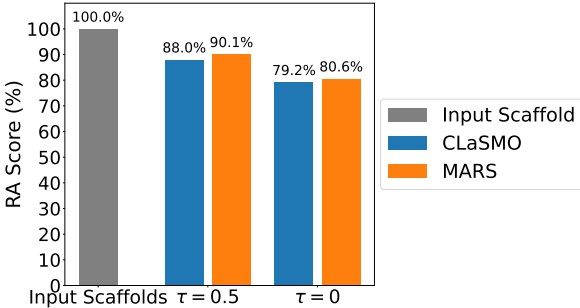

Figure 10: RA scores for input scaffolds and molecules generated by CLaSMO and MARS. Bars indicate the percentage of molecules predicted to be synthesizable by a retrosynthesis planning model.

## 5.3 Ablation Study

To assess the individual contributions of each component in the CLaSMO framework, we conducted ablation studies on (i) the role of LSBO as the search strategy and (ii) the effect of conditioning on atomic environment features in the generative model. All evaluations were conducted in a 2D latent space for consistency and under a high similarity constraint ($\tau = 0.5$).

In the first setting, we use a standard VAE without conditioning, paired with LSBO, denoted as LSBO+VAE. In the second, we use the full CVAE model but replace LSBO with random search (RS), denoted as RS+CVAE. These are compared against the original LSBO+CVAE configuration used in CLaSMO. To evaluate these alternatives, we report the final Top-10 average scores across three representative oracle optimization tasks: mestranol_similarity, QED, and troglitazone_rediscovery.

As shown in Table 1, LSBO+CVAE achieves the highest performance, highlighting the synergy between guided search and conditional generation. LSBO+VAE underperforms compared the other methods, indicating that lack of conditioning leads to chemically incompatible fragments that are penalized under high similarity constraints. While LSBO provides sample-efficient search, it cannot overcome this semantic mismatch. RS+CVAE outperforms LSBO+VAE in two out of three tasks, demonstrating that conditioning helps narrow the search space to chemically meaningful and similarity-compliant substructures—even when sampled randomly. These results further underscores the importance of scaffold-aware conditioning in constrained optimization settings.

Together, these findings confirm that both components, conditional generation and LSBO, are essential for achieving high performance.

Table 1: QED optimization results from the ablation study. We compare three configurations: LSBO+CVAE (original setting), LSBO+VAE (without conditioning), and RS+CVAE (without guided optimization). Results are reported as Top-10 average QED scores under a similarity constraint of $\tau = 0.5$. The full CLaSMO configuration achieves the highest performance.

| Oracle | LSBO+CVAE | RS+CVAE | LSBO+VAE |
|---|---|---|---|
| mestranol_similarity | 0.331 | 0.303 | 0.308 |
| QED | 0.930 | 0.922 | 0.897 |
| troglitazone_rediscovery | 0.239 | 0.216 | 0.191 |

## 5.4 Interactive Optimization

In CLaSMO, the modification region of the input scaffold is typically selected during the automated optimization process. However, the framework also supports an interactive mode, where a chemical expert manually selects the region of the molecule to modify. In this mode, the expert identifies the specific atom or region for modification, rather than relying on CLaSMO's automated selection. Once the region is cho-

sen, the rest of the process remains unchanged—CLaSMO continues to optimize the molecule by generating substructures using the CVAE and refining them through LSBO to improve target properties.

This interactive approach offers several key advantages. It enables the integration of expert knowledge into the optimization process, allowing CLaSMO to operate in a Human-in-the-loop setting. Experts can leverage their domain-specific insights to target specific regions they find promising, ensuring that modifications are not only data-driven but also aligned with scientific understanding. Meanwhile, CLaSMO maintains its efficient optimization process, using LSBO to enhance molecular properties and preserve sample-efficiency. Detailed user guidelines for this open-source web application are provided in the Appendix A.12.

## 6 Conclusion

In this paper, we introduced CLaSMO, a novel framework that combines CVAE and LSBO for scaffold-based molecular optimization. Our approach efficiently explores latent spaces to optimize molecular properties in wide variety of objectives, while maintaining structural similarity with the input scaffold to improve the chances of real-world viability of optimized molecules. By conditioning substructure generation on the atomic environment of the target region in the input molecule, CLaSMO generates chemically meaningful modifications. The experimental results—encompassing tasks such as rediscovery of molecules and multi-property optimization settings—demonstrate that CLaSMO consistently delivers superior performance across diverse settings and optimization objectives of varying complexity. By utilizing a pioneering LSBO-based molecular editing framework that exhibits superior sample-efficiency, CLaSMO proves to be a highly effective methodology for low-budget molecular optimization scenarios. Furthermore, CLaSMO's ability to control structural divergence via similarity constraints enables consistent performance across diverse optimization tasks, while also yielding favorable outcomes in terms of synthetic accessibility. Although this paper focuses on scaffold-based modifications, CLaSMO is fully compatible with whole molecules, requiring no changes to its methodology. Additionally, we have open-sourced a web application to allow chemical experts to use CLaSMO in a Human-in-the-Loop setting, further extending its practical applicability. Overall, CLaSMO exemplifies the power of combining scaffold-based strategies with LSBO, offering a highly effective tool for targeted drug discovery and broader molecular design challenges.

**Limitations and Scope**

As discussed throughout this manuscript, CLaSMO is particularly well-suited for low-budget optimization scenarios where sample-efficiency is critical. It is inherently designed to perform additive modifications on a given scaffold, which gradually increases the molecular size and weight with each editing step. By operating on scaffolds, CLaSMO helps limit the extent of molecular growth, ensuring that the optimized molecules do not become too large compared to their original counterparts, with the help of its constrained optimization capabilities. However, the current framework does not support the removal or replacement of existing substructures, limiting its flexibility in certain optimization settings. CLaSMO may not be ideal in scenarios where maintaining a fixed molecular size is crucial, or where more flexible transformations—such as fragment replacement or crossovers between molecular series—are required. In such settings, its utility may be limited without integration into broader pipelines. Besides, with its emphasis on introducing small modifications to the input scaffolds and given the scope of this study, CLaSMO uses a set of small fragments in its training dataset. Depending on the requirements of the problem, training on more complex fragments may be necessary, as discussed in Appendix A.2.

Another limitation, related to the discussion above, is that CLaSMO is not suitable for de novo, from-scratch molecule generation. While it is effective at adding substructures to a given molecule, it is not designed to construct molecules from scratch. In principle, CLaSMO could be adapted to grow a molecule starting from a single atom or fragment by iteratively generating substructures, but such usage falls outside the intended design and is beyond the scope of this work.

Nonetheless, CLaSMO remains a valuable tool when scaffold-based additions are permissible. It can be effectively combined with other methods presented in this study, serving as a component within a modular molecular optimization framework tailored to diverse design constraints.

**Broader Impact Statement**

The work presented in this paper has the potential to accelerate the discovery of new chemical compounds, which can positively impact various industries, particularly pharmaceuticals and materials science. By improving the efficiency of molecular optimization, CLaSMO could contribute to the development of more effective drugs, especially in regions facing significant public health challenges, such as the need for rapid vaccine development. Moreover, CLaSMO's focus on real-world applicability increases the chances that the compounds discovered are not just theoretical but can be realistically produced, which is critical for translating scientific innovation into real-world solutions. Moreover, CLaSMO's ability to work in a Human-in-the-Loop setting enables domain experts to directly contribute to the optimization process, enhancing collaboration between artificial intelligence and human expertise.

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

# A  Appendix

## A.1  Definitions of Atomic Features

In Table 2, we provide the definitions of the six atomic features we utilized in our data preparation setting.

| Property | Description |
|---|---|
| Atom Type | Specifies the element (e.g., carbon, oxygen), which determines bonding capabilities and chemical behavior. |
| Hybridization | Describes the mixing of atomic orbitals, influencing shape, bond angles, and bonding interactions. |
| Valence | Refers to the number of bonds an atom can form, indicating potential for additional bonding. |
| Formal Charge | Represents the charge if all bonding electrons are shared equally, crucial for reactivity and bonding sites. |
| Degree | Denotes the number of directly attached atoms (neighbors), providing insight into the local atomic environment. |
| Ring Membership | Indicates if the atom is part of a ring structure, impacting substructure rigidity, stability, and bonding behavior. |

Table 2: Key atomic properties used in CVAE training in CLaSMO framework.

## A.2  Training with BRICS Fragments from the ZINC250K Dataset

Our initial decision to use BRICS fragments derived from the QM9 dataset was motivated by our goal of training a generative model capable of making small, chemically interpretable modifications to input

scaffolds. However, the CLaSMO framework is not limited to the use of QM9. Depending on the task at hand, the CVAE can be trained on BRICS fragments from alternative datasets that offer different levels of structural complexity and diversity.

As discussed in Section 5.1.1, BRICS fragments obtained from ZINC250K exhibit a wider range of sizes and chemical diversity compared to those from QM9. In this section, through simple experiments, we investigate the implications of using the ZINC250K dataset within the CLaSMO framework.

### A.2.1 Training CVAE with Only ZINC250K Fragments

We first conduct a preliminary study in which we train the CVAE model using the complete set of BRICS fragments extracted from ZINC250K, without applying any molecular weight filtering. This results in 35,232 unique BRICS fragments and their corresponding atomic environment features, significantly increasing the chemical diversity and structural complexity available during training.

To assess the effect of this increased diversity and fragment size, we trained CVAE models using this ZINC250K-derived dataset with latent dimensions of 2 and 8, keeping the model architecture, hyperparameters, and training procedure consistent with the QM9-based setting. The model with a 2-dimensional latent space achieved a test reconstruction accuracy of 87.1%, while the model with an 8-dimensional latent space achieved 91.2%. Despite the added complexity of the ZINC250K fragments, the 2D model still performed reasonably well, though further improvements could likely be achieved with proper hyperparameter tuning.

We then evaluated the property optimization performance of the 2D ZINC250K-trained model in a QED optimization task and benchmarked it against the original CLaSMO model trained on QM9-derived fragments. We also compared the average molecular weight of the optimized molecules.

As shown in Table 3, the use of ZINC250K-derived fragments results in the generation of molecules with substantially higher molecular weights—averaging 473.12 Da—due to the inherently larger size of the training fragments. Under the $\tau = 0$ similarity setting, the ZINC250K-trained model achieves a top-10 average QED of 0.890. This performance surpasses all baseline methods except MARS, which it closely approaches, demonstrating that the model remains effective even when operating with a more structurally complex fragment vocabulary. In the following subsection (Appendix A.2.2), we show that using small fragments filtered from the ZINC250K dataset in combination with the QM9 dataset is more suitable for our model, which consistently outperforms the baselines, including MARS.

However, under the more restrictive $\tau = 0.5$ setting, the model's performance drops more noticeably, achieving a QED of only 0.871. This decline can be attributed to the larger substructures generated by the model, which make it difficult to maintain high similarity to the input scaffold. The model struggles to produce small, localized modifications, which are essential under stricter similarity constraints. In contrast, the original QM9-based CLaSMO setting maintains high performance across both similarity thresholds, highlighting the importance of fragment size in preserving molecular similarity during optimization.

It is also important to note that the current model architecture was originally designed for QM9-sized fragments and has not been adapted to handle the increased complexity introduced by the ZINC250K fragments, many of which exceed 400 Da in molecular weight. With a more carefully selected model architecture and appropriate hyperparameter tuning—such as latent dimensionality, condition embedding size, and decoder capacity—we expect this setup to yield stronger performance. In the next substructure, we show a more suitable alternative to scale CLaSMO with new fragments.

### A.2.2 Training CVAE with a Mixture of ZINC250K and QM9 Fragments

As discussed in Section 5.1.1, the ZINC250K dataset contains some BRICS fragments that are comparable in size to those extracted from QM9. By filtering for such fragments, we curated a subset of 4,706 unique BRICS fragments along with their corresponding atomic environment vectors. This subset enhances the chemical diversity of the training data while maintaining the small fragment size necessary for fine-grained scaffold editing.

Table 3: Comparison of average optimized molecular weight (Avg. Mol. Weight) and top-10 average QED scores for models trained on BRICS fragments from different datasets.

| CLaSMO Setting | Avg. Mol. Weight $\tau = 0$ | QED $\tau = 0$ | QED $\tau = 0.50$ |
|---|---|---|---|
| ZINC250K Fragments, LD:2 | 473.12 | 0.890 | 0.871 |
| QM9 Fragments, LD:2 (Original Setting) | 320.08 | 0.932 | 0.930 |
| MARS | 313.28 | 0.896 | 0.890 |

Table 4: Comparison of CLaSMO performance across three tasks—QED optimization, troglitazone_rediscovery, and mestranol_similarity—under different fragment settings and similarity constraint $\tau = 0$. The setting that combines QM9 and ZINC250K fragments demonstrates comparable performance to the original QM9-only configuration and outperforms the MARS baseline across all tasks, highlighting CLaSMO's robustness and potential for scaling with diverse fragment sources.

| CLaSMO Setting | QED | troglitazone_rediscovery | mestranol_similarity |
|---|---|---|---|
| QM9 + ZINC250K Fragments, LD:2 | 0.912 | 0.233 | 0.342 |
| QM9 Fragments, LD:2 (Original Setting) | 0.932 | 0.243 | 0.355 |
| MARS | 0.896 | 0.216 | 0.286 |

We augmented the QM9-derived dataset with a filtered subset of BRICS fragments from ZINC250K and trained a CVAE model using a 2-dimensional latent space, consistent with the original setup. No additional hyperparameter tuning was performed; all settings from the original CLaSMO configuration were reused for both model training and the LSBO procedure.

In the QED optimization task, the mixed-fragment model achieved a Top-10 average QED of 0.912—surpassing all baseline models and closely approaching the performance of the original CLaSMO setup. The average molecular weight of the generated molecules was 327.04, remaining aligned with the QM9-only setting. We further evaluated this configuration on the troglitazone_rediscovery and mestranol_similarity benchmarks with no similarity constraint ($\tau = 0$). As shown in Table 4, the model outperformed the strongest baseline method, MARS, and performed comparably to the original CLaSMO setting across all metrics, confirming the method's robustness to variation in fragment sources.

These results highlight CLaSMO's strong generalization capabilities, even when exposed to more structurally diverse training data. Notably, the model achieved this level of performance without any additional tuning, suggesting that task-specific or dataset-aware hyperparameter optimization could lead to further improvements. This demonstrates CLaSMO's potential to scale effectively with alternative datasets composed of small molecular fragments.

### A.3 Model and Hyperparameter Selection

In chemical VAE models, it has been established that setting the KL divergence weight $\beta < 1$ can improve generative performance Yan et al. (2020), and our findings are consistent with this. We experimented with a range of $\beta$ values from 1 to $1^{-7}$, selecting models based on their reconstruction accuracy on the training set. Interestingly, we found that even at very low $\beta$ values, the CVAE retained its generative capacity. However, as $\beta$ increased, we observed a decline in both reconstruction accuracy and the diversity of generated molecules.

In chemical VAE models, it has been established that setting the KL divergence weight $\beta < 1$ can improve generative performance Yan et al. (2020), and our findings are consistent with this. We experimented with a range of $\beta$ values from 1 to $1^{-7}$, selecting models based on their reconstruction accuracy on the training set. Interestingly, we found that even at very low $\beta$ values, the CVAE retained its generative capacity. However, as $\beta$ increased, we observed a decline in both reconstruction accuracy and the diversity of generated molecules. To ensure robust training, we employed early stopping in conjunction with a learning rate scheduler. Specifically, we used PyTorch's ReduceLROnPlateau, which reduces the learning rate by a factor of 0.5 if the validation loss does not improve for 5 consecutive epochs. Early stopping was triggered

if no improvement was observed within 10 epochs. The initial learning rate was set to 0.001. Models were evaluated based on their reconstruction performance and generative diversity, leading us to select the model with $\beta = 0.000001$ as the optimal candidate. Additionally, our CVAE leverages conditional batch normalization De Vries et al. (2017), which improves the impact of conditioning in the generative process.

For the kernel used in CLaSMO, the lengthscale parameters of both the RBF and categorical kernels are learned by maximizing the log marginal likelihood during GP training. In our mixed-variable optimization setup, the continuous latent vectors and discrete attachment point (atom id) indices are treated jointly. During acquisition function optimization, the categorical variables (attachment point indices) are temporarily relaxed to continuous values. This allows the use of gradient-based optimization across the full search space. After optimization, the relaxed categorical dimensions are rounded to the nearest valid integer index. This approach enables efficient end-to-end optimization while avoiding the computational cost of enumerating over all discrete options.A similar strategy for handling categorical and integer-valued variables in BO is discussed in prior work by Garrido-Merchán & Hernández-Lobato (2020).

For CLaSMO's penalization terms, we opted for a straightforward approach rather than an exhaustive hyperparameter optimization process. We set $\lambda_1 = -5$ to penalize cases where the similarity constraint $\text{DICE}(\boldsymbol{S}, \boldsymbol{S'}) > \tau$ was violated. Similarly, we assigned $\lambda_2 = -7.5$ for situations where the generated substructure could not be added to the input scaffold. These values were chosen to assign poor scores in cases where the sampled region did not meet the desired criteria, allowing LSBO to learn that the region is suboptimal. This approach helps guide the optimization process away from unproductive regions and toward more promising areas.

### A.4 Condition Vector Embeddings

The simplicity of our dataset allowed us to reduce the latent space to two dimensions without compromising reconstruction quality, thereby enhancing LSBO performance. Consequently, the decision to use 2-dimensional embeddings for the condition vectors, rather than the original six-dimensional features, was motivated by this low-dimensional latent space configuration in our CVAE model.

Using a six-dimensional condition vector in a CVAE with a 2-dimensional latent space poses a significant challenge: the decoder may overly rely on the condition vectors during reconstruction, potentially diminishing the latent space's ability to encode critical structural information. This can result in a loss of critical structural information within the latent space, as a disproportionate amount of information is drawn from the condition vectors. Such an imbalance can degrade the quality of the latent space, negatively affecting the surrogate model used in LSBO and ultimately hindering the optimization process. Conversely, when 2-dimensional condition vectors are used, the latent vectors must encode more information about the input instances, as the decoder's reliance on the condition vectors is reduced. This shift encourages the latent space to better capture the underlying structural properties, improving its utility for LSBO. In LSBO, latent vectors are utilized to train the surrogate model, which guides the search process by predicting target property values within specific regions of the latent space. Therefore, ensuring that the latent vectors are information-rich is crucial for achieving efficient LSBO performance.

To evaluate this effect, we compared CVAE models trained with 2-dimensional embeddings of the conditional features to those trained with the original 6-dimensional features. For both models, we obtained the 2-dimensional latent vectors of input scaffolds using the encoders of the trained CVAEs and calculated the corresponding QED values. We then trained random forest regressors to predict the QED values of the input scaffolds based on the latent vectors from each model. Our results showed that the CVAE with 2-dimensional condition embeddings achieved a 5% lower mean squared error in QED predictions, underscoring the benefits of reduced-dimensional embeddings in preserving latent space quality and improving LSBO performance.

We acknowledge that the model with 6-dimensional condition vectors converged faster and achieved slightly better reconstruction accuracy during its training compared to model with 2-dimensional condition vector embeddings. However, our primary goal was to design the latent space for LSBO, where the latent representation provided by the 2-dimensional embeddings proved more effective.

## A.5 Input Scaffold Characteristics

In our study, in order to use scaffolds of different sizes and different chemical varieties, we used 100 scaffolds sampled from QM9 and ZINC250 datasets. Namely, we used 37 scaffolds from QM9 and 63 scaffolds from ZINC250K datasets. In Figure 11, we provide the distribution of molecular weights of the input scaffolds along with the distributions of the 20 properties of these scaffolds that are considered in our molecular optimization experiments. Distributions demonstrate that these scaffolds are fairly diverse in many of these 21 distributions. On the other hand, in Fig. 12, we provide the full list of the visualization of the input scaffolds, which ranges from small scaffolds to bigger ones.

## A.6 Sample-Efficiency Benchmark Experimental Setting

In the sample-efficiency benchmark experiments adopted from Gao et al. (2022), we reduced the number of optimization iterations from 10,000 to 100 to create a low-budget experimental setting. For each methodology we limited the number of oracle calls to 100.

In the original open-source implementation, Smiles-GA uses a population size of 50, while Graph-GA uses a population size of 120. Given our limited optimization budget of only 100 iterations, these values required adjustment. Furthermore, Smiles-GA allows for 500 mutations in its default configuration, and Graph-GA employs a mutation rate of 6.7%. As we report results on the cases where the scaffold is preserved, we looked for the balance between scaffold preservation and exploration. For this reason, we conducted a grid search for selecting the best-performing sets of population size and mutation rate for the Graph-GA methodology. Population size is selected among $4, 5, 10$ and the candidate set of mutation rate is $6.7\%, 4.5\%, 1\%, 0.1\%, 0.05\%, 0.01\%, 0.005\%$. The best results were obtained when the mutation rate is equal to 4.5% and the population size is equal to 5. For SMILES-GA, we experimented with population sizes of $4, 5, 10$ by setting the number of mutations to 10, and best results were obtained when the population size was set to 5.

In the case of SynFlowNet, the open-source implementation provided by the authors was not directly applicable to scaffold optimization tasks. However, the authors kindly guided us on how to adapt their approach for this setting, and we updated their codebase accordingly based on their recommendations.

The Stoned methodology, although categorized as a genetic algorithm in Gao et al. (2022), relies on random rearrangements of atoms and lacks a specific hyperparameter to control the degree of divergence from the initial scaffold. Consequently, similar to MARS, and MolDQN, we used the default hyperparameters for the experiments using Stoned methodology to optimize the given scaffolds.

## A.7 Oracle Definitions

In this section, we provide brief descriptions of the oracle functions used in our experiments. For additional details and references, we refer the reader to the Therapeutics Data Commons website: `https://tdcommons.ai/functions/oracles/`.

**Group 1: Similarity and Rediscovery Tasks**

- **albuterol_similarity:** Measures similarity to albuterol, a bronchodilator.

- **celecoxib_rediscovery:** Regenerates celecoxib, an anti-inflammatory drug.

- **mestranol_similarity:** Quantifies similarity to mestranol, a synthetic estrogen.

- **thiothixene_rediscovery:** Seeks to recover thiothixene, an antipsychotic.

- **troglitazone_rediscovery:** Rediscovery task for troglitazone, a former diabetes drug.

**Group 2: Multi-Property Optimization (MPO) Tasks**

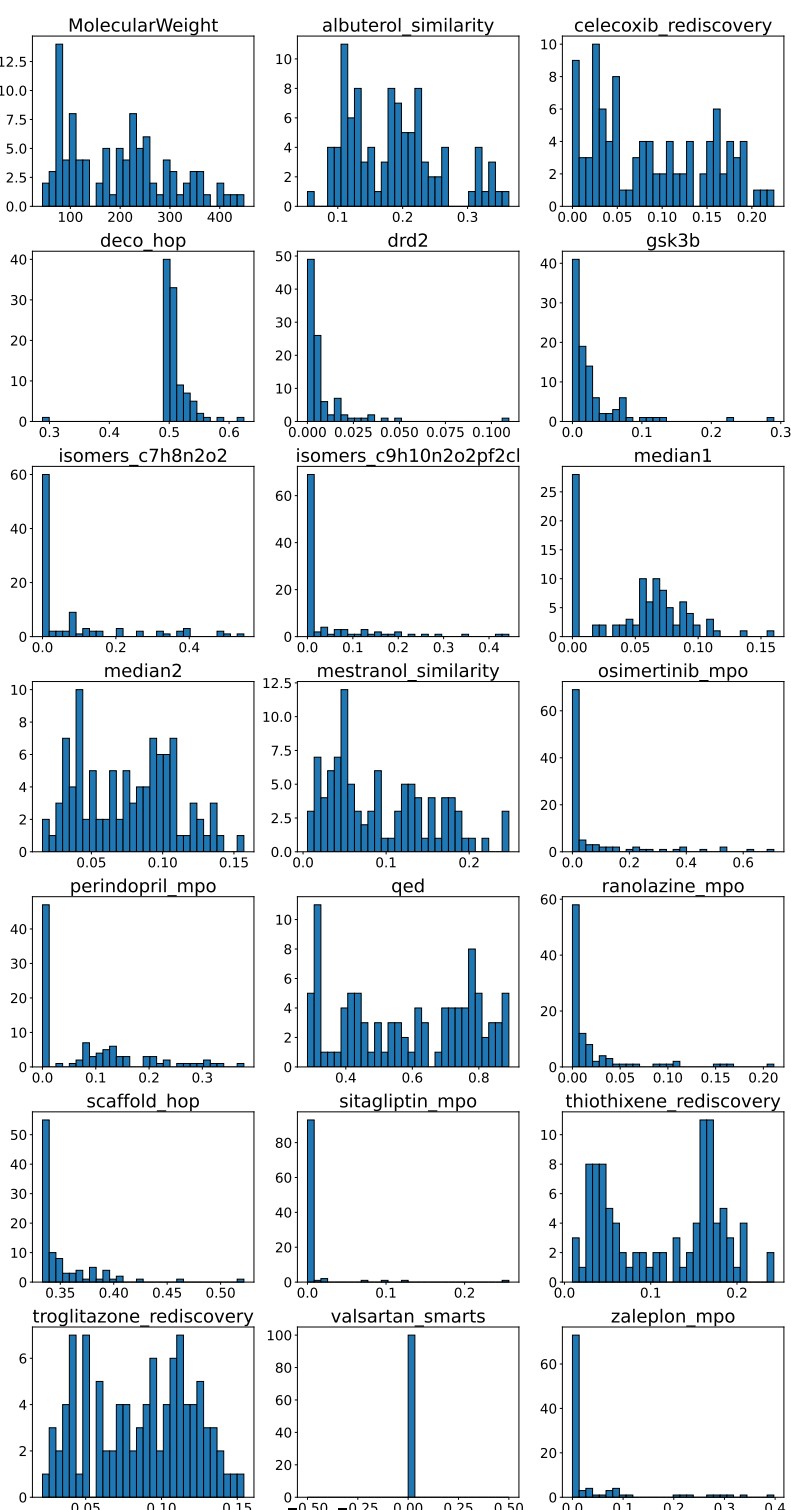

Figure 11: Distribution of molecular weights and the property values of the input scaffolds.

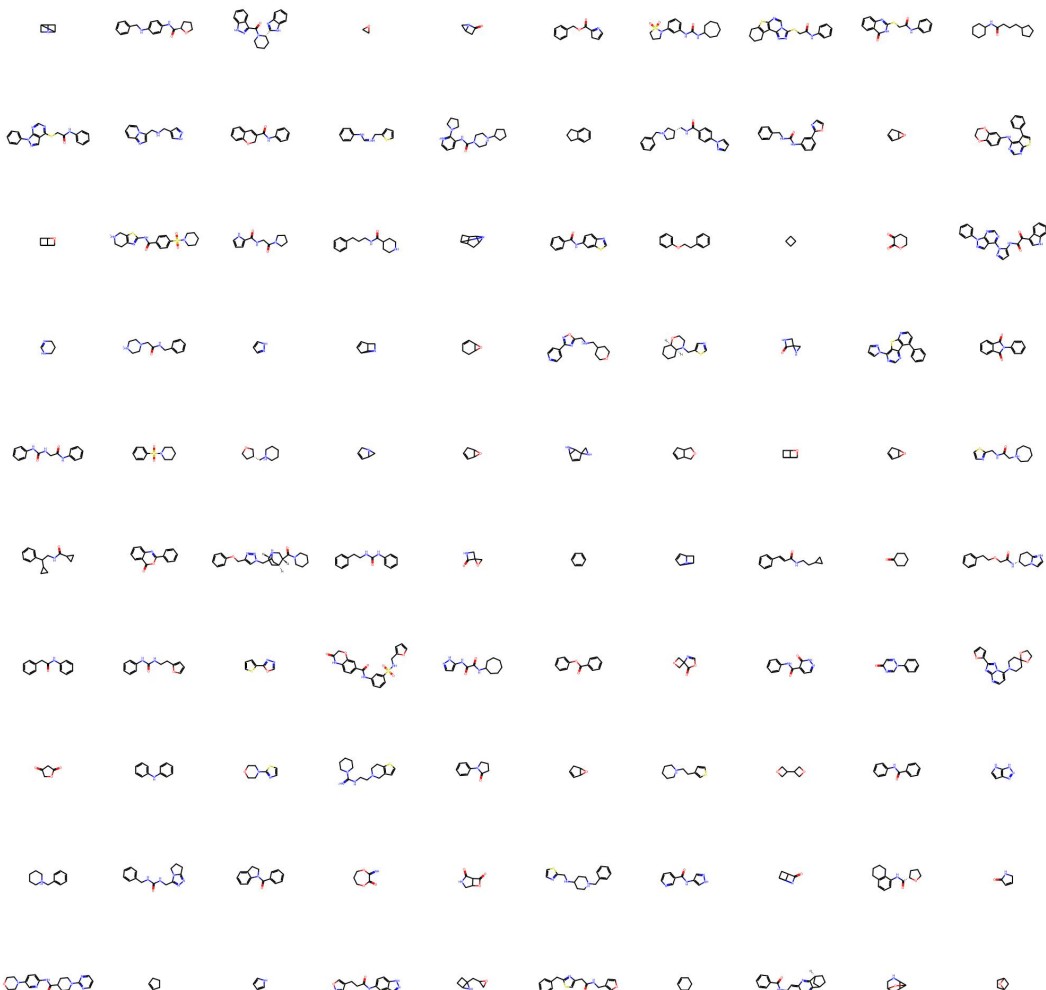

Figure 12: Visulizations of input scaffolds.

- **osimertinib_mpo:** Multi-property optimization for osimertinib, an EGFR inhibitor.

- **perindopril_mpo:** Focused on perindopril, an ACE inhibitor.

- **ranolazine_mpo:** Related to ranolazine, an anti-anginal compound.

- **sitagliptin_mpo:** MPO task for sitagliptin, an antidiabetic drug.

- **zaleplon_mpo:** Targets properties associated with zaleplon, a sedative-hypnotic.

**Group 3: Structural Modification Tasks**

- **deco_hop:** Scaffold hopping task with partial structural retention.

- **scaffold_hop:** Generates novel scaffolds while maintaining core activity.

- **isomers_c7h8n2o2:** Generates valid isomers with the formula $C_7H_8N_2O_2$.

- **isomers_c9h10n2o2pf2cl:** Generates valid isomers of $C_9H_{10}N_2O_2PF_2Cl$.

**Group 4: Activity and Benchmark Tasks**

- **drd2:** Predicts activity against dopamine receptor D2.

- **gsk3b:** Predicts activity against GSK3$\beta$, a kinase target.

- **median1, median2:** Benchmark tasks requiring similarity across multiple target molecules.

- **qed:** Quantitative Estimate of Drug-likeness.

- **valsartan_smarts:** Checks for specific SMARTS patterns associated with valsartan.

### A.8 Similarity Constraints Evaluation

In our experimental framework, molecular similarity constraints are defined using the Dice similarity between the input scaffold and the optimized molecule. These constraints are designed to reflect varying levels of tolerance for structural modifications, which are common in real-world molecular design scenarios. However, it is important to note that there is no universally accepted threshold for what constitutes a "highly similar" molecule, as similarity interpretations can vary depending on the fingerprinting method and application context.

To determine suitable threshold values, we conducted an empirical analysis of similarity trends observed during the iterative modification of molecules. In particular, we examined the relationship between similarity and factors such as the number of added fragments, their molecular weights, and the size of the original scaffold. Based on these observations, we designate a Dice similarity threshold of 0.5 as the high-similarity constraint, as this value permits limited modifications to smaller input scaffolds while still keeping the optimized molecule relatively close to the input scaffold when larger scaffolds are used. Thresholds of 0.25 and 0 are adopted to represent moderate and no similarity constraints, respectively, allowing for increasing flexibility in structural modifications.

Figure 13 illustrates the evolution of both Dice and Tanimoto similarity values as the input scaffold is sequentially modified through fragment additions. As shown, Dice similarity typically falls below 0.5 after only a few additions. This behavior is strongly influenced by the size of the input scaffold: smaller scaffolds exhibit a more rapid decline in similarity, as each modification introduces a relatively larger structural change. In contrast, larger scaffolds can accommodate multiple modifications before crossing the same threshold. As detailed in Appendix A.5, our dataset contains both small and large scaffolds. This analysis reveals that applying a strict similarity constraint disproportionately restricts modifications in smaller scaffolds, whereas a more relaxed constraint enables at least one meaningful modification across a broader range of scaffold sizes. We found that setting the high similarity threshold to 0.50 provides a good trade-off: it permits a single modification in the smallest scaffolds while maintaining a desirable level of similarity for larger ones.

Figure 13: Illustration of the change in Dice and Tanimoto similarities as the input scaffold gets modified. After a few additions, dice similarity goes below 0.5.

### A.9  Analysis on Bonding Success During LSBO

An important metric for evaluating the effectiveness of CLaSMO is the success rate of bonding between the generated substructures and the original scaffold. In each optimization run, CLaSMO is executed for 100 iterations, and the computational budget is consumed regardless of whether bonding is successful. Therefore, it is crucial to ensure that our condition-based generation strategy consistently produces substructures capable of forming valid bonds with the scaffold—especially in cases where the scaffold contains atoms with available bonding capacity.

In Table 5, we present an analysis of bonding success across all 20 oracle optimization tasks when the similarity constraint $\tau$ is set to 0, averaged over all seeds. We report the proportion of successful bonds formed within the first 10, 25, 50, 75, and all 100 iterations. Our findings show that CLaSMO achieves more than 99% bonding success within the first 10 iterations, and over 98% within the first 25 iterations across all oracle tasks. These results demonstrate the high reliability of the model in generating chemically compatible substructures early in the optimization process.

As outlined in Algorithm 1, the molecule is only updated when a generated substructure improves the target property. As optimization progresses and the target property increases, achieving further improvement becomes more difficult. Consequently, LSBO is pushed to explore more sparse and uncertain regions of the latent space in search of viable candidates. This behavior is especially evident in the latter half of the optimization process. While the success rate of bonding remains above 80% for all oracle tasks up to iteration 50, it begins to decline thereafter. This drop can be attributed both to increased exploration and to a reduction in the number of atoms within the scaffold that are still capable of forming additional bonds. As a result, successfully generating and bonding new substructures becomes progressively more challenging in later stages.

### A.10  More on MARS and CLaSMO Performances

Across the three similarity constraint settings—no constraint, moderate ($\tau = 0.25$), and high similarity ($\tau = 0.50$)—CLaSMO consistently demonstrates superior optimization performance, ranking first among 13 methods under both moderate and high similarity conditions, and first among 11 methods in the unconstrained setting. This highlights its robustness and versatility across a wide range of molecular design scenarios. In contrast, MARS exhibits strong performance in multi-property optimization tasks, particularly when no similarity constraint is imposed. This suggests that MARS is well-suited for exploring broader chemical

Table 5: Bonding success rate of generated substructures during LSBO across 20 oracle optimization tasks. The table reports the percentage of substructures that successfully formed a valid bond with the input scaffold within the first 10, 25, 50, 75, and 100 iterations. The results demonstrate high bonding reliability in early iterations, with a gradual decline as the search space becomes more sparse and scaffold bonding capacity diminishes.

| Property Name | Iter 10 | Iter 25 | Iter 50 | Iter 75 | Iter 100 |
|---|---|---|---|---|---|
| albuterol_similarity | 99.9% | 99.6% | 94.3% | 81.3% | 64.8% |
| celecoxib_rediscovery | 100% | 100% | 98.0% | 90.5% | 74.9% |
| deco_hop | 100% | 98.6% | 80.5% | 58.1% | 44.1% |
| drd2 | 100% | 99.8% | 94.5% | 81.2% | 66.2% |
| gsk3b | 100% | 99.8% | 94.6% | 81.2% | 66.3% |
| isomers_c7h8n2o2 | 99.9% | 99.8% | 97.4% | 85.0% | 67.4% |
| isomers_c9h10n2o2pf2cl | 100% | 99.9% | 97.4% | 87.4% | 69.3% |
| median1 | 100% | 99.9% | 97.5% | 89.4% | 75.0% |
| median2 | 100% | 100% | 98.4% | 90.4% | 76.8% |
| mestranol_similarity | 100% | 100% | 96.7% | 80.3% | 61.4% |
| osimertinib_mpo | 99.8% | 98.2% | 82.8% | 61.6% | 46.5% |
| perindopril_mpo | 100% | 100% | 98.1% | 86.9% | 69.4% |
| qed | 99.4% | 98.1% | 89.2% | 70.8% | 53.9% |
| ranolazine_mpo | 100% | 99.8% | 90.3% | 66.0% | 49.6% |
| scaffold_hop | 100% | 98.6% | 81.2% | 58.6% | 44.3% |
| sitagliptin_mpo | 99.9% | 99.4% | 94.5% | 77.7% | 59.9% |
| thiothixene_rediscovery | 100% | 100% | 99.0% | 93.4% | 80.6% |
| troglitazone_rediscovery | 100% | 99.9% | 95.9% | 84.0% | 68.5% |
| valsartan_smarts | 100% | 100% | 100% | 99.1% | 87.3% |
| zaleplon_mpo | 100% | 99.7% | 94.3% | 75.2% | 57.2% |

spaces where structural similarity to the starting molecule is less critical. As discussed in Section 5.2.3, the SA and RA scores of the molecules generated by both methods are highly comparable, indicating that each approach is capable of producing molecules with high synthesizability. Taken together, these findings suggest that MARS may be the preferred method for multi-property optimization when similarity constraints can be relaxed, whereas CLaSMO is better suited for scenarios requiring high sample-efficiency, strict structural similarity, or focused single-property optimization.

### A.11 More on Synthetic Accessibility Scores

Conducting a fair comparison of synthetic accessibility (SA) scores requires focusing on settings where meaningful optimization has occurred. To this end, we included only those experimental configurations in which at least 30 out of 100 input scaffolds were successfully optimized—that is, they resulted in an improvement in the target oracle value. This criterion ensures that the SA score analysis reflects the overall synthesizability of molecules that were effectively optimized.

### A.12 Human-in-the-Loop via Web-Application

In Fig. 14, we showcase a sequence of screenshots from our web application, demonstrating the process of molecule optimization. First, the user inputs a SMILES Weininger (1988) string of the chemical compound into the designated text field. Once the input is provided, the application automatically computes the molecule's QED value and generates a visual representation. Subsequently, the user selects a region of interest by drawing a rectangle around the area they wish to modify. Upon confirming the selection, the CLaSMO optimization process is initiated, targeting improvements in the selected molecular region. Upon completion, the optimized molecule is displayed, and the process can be continued by using the resulting molecule as input for further iterations. By incorporating user input in the region selection, we create a Human-in-the-Loop optimization workflow.

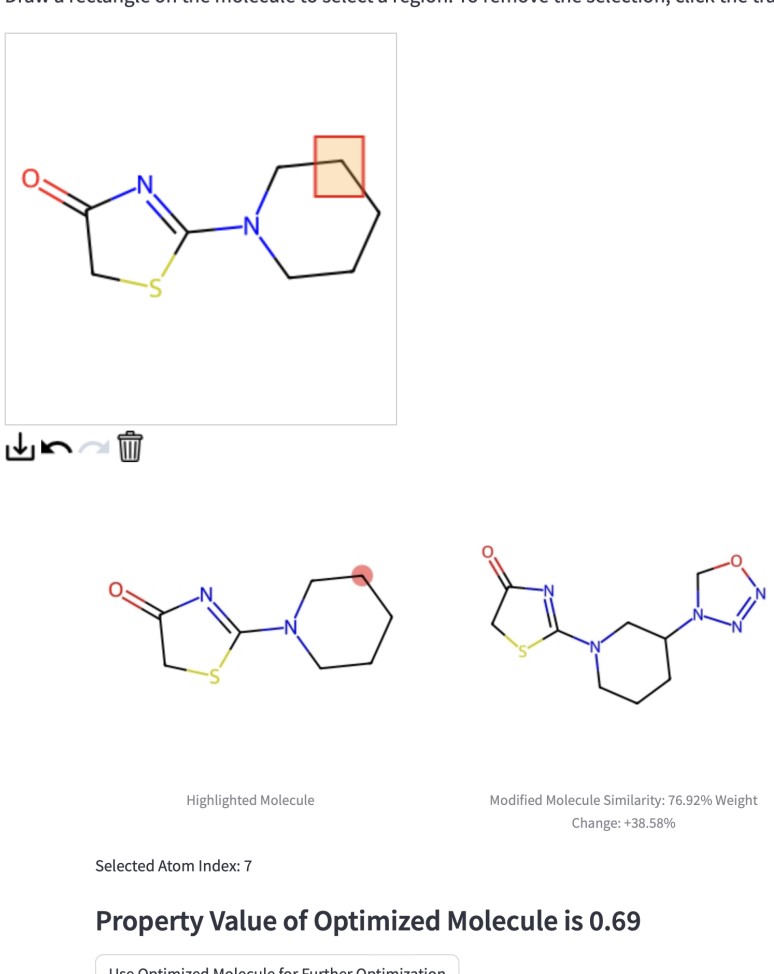

Figure 14: Screenshots from our web application showing the step-by-step process of molecule optimization using CLaSMO in an interactive setting. The process includes inputting a SMILES string, visualizing the molecule's QED value, selecting a region for modification, and initiating the optimization procedure. In this example, the QED value of the input molecule is improved from 0.56 to 0.69, where the resulting molecule is demonstrated in the bottom-right figure.

