# OpenReview forum: "Conditional Latent Space Molecular Scaffold Optimization for Accelerated Molecular Design"
_TMLR — Accepted by TMLR_

### Review · Reviewer_pbL8 · 2025-06-11

**Summary Of Contributions:**

This work proposes a method for optimizing the properties of small molecules, conditioned on a fixed molecular scaffold. Specifically, the decoder of a Conditional Variational Autoencoder is employed to sample molecular substructures that can potentially be attached to the scaffold at designated attachment points. The optimization process — i.e., identifying a latent variable and an attachment point that together yield a beneficial substructure — is guided by Bayesian optimization, using Gaussian Processes with an Upper Confidence Bound acquisition function. The method is evaluated on 20 molecular optimization tasks, where fitness is assessed via oracle functions. Across these tasks, the proposed approach consistently outperforms the baseline methods.

**Audience:**

Yes

**Broader Impact Concerns:**

no concerns.

**Claims And Evidence:**

Yes

**Requested Changes:**

(Q1 – CVAE Latent Space Capacity):
The authors report a test-set reconstruction accuracy above 90%, despite using a CVAE with a two-dimensional latent space. In light of (W1), could this high performance be attributed to the simplicity of the molecules and substructures considered? How diverse and complex are the substructures used for training? It would be valuable to see the CVAE trained on a more chemically diverse subset, such as from ChEMBL, to assess whether the latent space can still support accurate reconstruction across a broader range of molecular structures.

(Q2 – Scaffold Diversity in Optimization):
I assume the optimization tasks are based solely on QM9-derived scaffolds—is this correct? If so, this significantly limits the experimental relevance, as it leaves open the question of whether the approach generalizes to more structurally diverse or drug-like scaffolds. The authors should clarify and explicitly report the source and diversity of the scaffolds used.

(Q3 – Conditioning on Attachment Points Only):
The substructure sampling appears to be conditioned only on the attachment point, not on the scaffold as a whole. Is there a specific rationale behind ignoring the scaffold context? If so, the authors should justify this design choice and discuss it in the context of related work. Intuitively, the scaffold might influence the compatibility and desirability of attached substructures, so this simplification deserves further explanation.

(Q4 – Comparison to Diffusion-Based Methods):
How do the authors anticipate diffusion-based models would perform in their benchmark tasks? For example, the recently proposed GenMol model [1] includes a section on goal-directed lead optimization. While [1] can be considered concurrent work and is thus not expected to be included in the experimental section, its relevance to the problem setting justifies its discussion in the manuscript.

### Reference:
[1] Lee et al., GenMol: A Drug Discovery Generalist with Discrete Diffusion, 2025.

**Strengths And Weaknesses:**

The proposed method is interesting and relevant (see S1 – S3). However, it remains unclear whether it is applicable only to relatively simple scenarios (QM9-like molecules), or whether it can scale to more complex cases involving more structurally diverse molecular scaffolds.

### Strengths
(S1 – Relevance): Optimizing a molecule's property while making minimal structural changes is a highly relevant problem in drug discovery. Solutions addressing this challenge are therefore inherently valuable.

(S2 – Clarity & Manuscript Quality): The manuscript is clearly written and presents a well-defined problem and research objective. The proposed method is described with clarity and is well supported by the introduction, background, and a thoughtfully selected review of related work.

(S3 – Significance): The proposed approach consistently outperforms baseline methods across a diverse set of optimization tasks.

### Weaknesses
(W1 – Potentially Oversimplified CVAE Training Setup): It is unclear why the QM9 dataset was chosen to train the CVAE, given that it only includes a narrow subset of chemical structures (molecules containing only H, C, N, O, and F). This choice limits molecular diversity and may restrict the broader applicability of the approach. A more chemically diverse dataset — such as a filtered subset of PubChem or ChEMBL — might have been more appropriate. Additionally, applying a small-molecule filter at 100 Daltons seems overly restrictive, considering that molecules up to 500 Daltons are typically considered small. Overall, the method appears to have been developed and evaluated in a constrained region of chemical space, potentially resulting in an overly simplified experimental setup (see also Q1, Q2). At present, it is unclear whether the proposed method can scale to more diverse molecular scaffolds (see W2).

(W2 – Limited Insight into CVAE Latent Space and Scalability): The experiments show that simple molecular structures from QM9 can be reconstructed using a low-dimensional (2D) latent space. However, it remains unclear whether a CVAE can effectively model and disentangle more complex molecular structures. This uncertainty limits confidence in the generalizability of the proposed approach. The authors should either demonstrate that their method can also be applied to more structurally complex molecules, or explicitly state that it is currently restricted to generating QM9-like molecules.

---

> ### Author Response · Authors · 2025-07-29
> **Updates on CVAE Training and Dataset Selection & CVAE Capacity & Scaffold Diversity (1/2)**
>
> We thank the reviewer for the detailed and constructive feedback, as well as for highlighting the relevance, clarity, and performance of our method. We also appreciate the critical points regarding dataset choice, latent space dimensionality, and scalability. Below, we respond to each concern and explain the corresponding updates made to the manuscript.
>
> ## W1 – Potentially Oversimplified CVAE Training Setup
>
> Thank you for this comment. We understand reviewers' concerns on dataset selection and its implications, and comments on the molecule filter at 100 Dalton. We would like to emphasize that our methodology is designed to add plausible and small substructures to given input molecules/scaffolds. In this sense, we do not impose any molecular weight limitations on the input molecules/scaffolds; they can be of any size and weight. The reason for introducing a molecular weight filter for the substructures to train our model is to make sure that our model generates small substructures, and adding these generated substructures to the input molecules does not result in a substantial change in the input molecule/scaffold to keep similarity to the input molecule at a high level while optimizing target properties.
>
> In the updated manuscript, we included a detailed discussion on our dataset selection process in Section 5.1.1. We demonstrate the BRICS fragments found from QM9 and ZINC250K dataset, and compare their molecular weights. As the ZINC250K dataset includes fragments that are heavy and large, and the QM9 dataset provides small fragments that are mostly less than a molecular weight of 100 Dalton, we selected the QM9 dataset for the sake of completeness and simplicity, and we find it sufficient for the experiments conducted in our study and for the scope of this work. However, one can, of course, filter smaller fragments from different publicly available datasets and use them to train our proposed method to introduce a larger variety. We dedicated a new section in the Appendix, Appendix A.2, for this alternative. In Section Appendix A.2.1, we evaluate the case where only ZINC250K fragments are used without any filtering. We show that this case fails especially when there are similarity constraints, as this model tends to generate larger substructures that introduce larger modifications to the input molecules. In Section Appendix A.2.2, we evaluate another alternative where smaller fragments from the ZINC250K dataset are used in addition to the original QM9 fragments. In this setting, we observe that it performs very closely to the original setting in QED, troglitazone_rediscovery, and mestranol_similarity optimization tasks, and outperforms other baselines. As these results are obtained without fine-tuning of the hyperparameters of this new model, we can expect it to perform better with careful tuning of its CVAE and LSBO-based hyperparameters, showcasing its potential to be scaled.
>
> ## W2 – Limited Insight into CVAE Latent Space and Scalability & Q1 – CVAE Latent Space Capacity
>
> Thank you for your comments. Since W2 and Q1 are closely related, we address them together.
>
> In the revised manuscript, we elaborate on our dataset selection process in Section 5.1.1, where we compare BRICS fragments derived from the QM9 and ZINC250K datasets. We chose QM9-derived fragments primarily because they are small, chemically consistent, and aligned with our modeling objective of performing minimal, context-aware modifications to molecular scaffolds. Given this scope, a 2-dimensional latent space proved sufficient to achieve high reconstruction accuracy (above 90\%), as noted by the reviewer.
>
> To evaluate scalability and address the question of generalization to more complex chemical structures, we conducted additional experiments presented in Appendix~A.2. In Appendix A.2.1, we trained the CVAE using only ZINC250K fragments. Without any hyperparameter tuning, the model achieved 87.1\% reconstruction accuracy with a 2-dimensional latent space and 91.2\% with an 8-dimensional latent space. As we discussed in our response to Weakness 1 (W1), we also evaluated these settings using property optimization experiments in Appendix A.2.1 and Appendix A.2.2, showcasing potential to scale beyond the QM9 dataset when small fragments from other datasets are used. However, if the problem setting requires the use of more complex molecules, it may also be useful to explore further hyperparameter or model architecture tuning.
>
> Overall, these findings support the flexibility of the CVAE architecture and indicate that the proposed method is not inherently limited to QM9-like molecules but can be extended to more diverse chemical spaces with appropriate model adaptations.

---

> > ### Author Response · Authors · 2025-07-29
> > **Updates on CVAE Training and Dataset Selection & CVAE Capacity & Scaffold Diversity (1/2)**
> >
> > ## Q2 – Scaffold Diversity in Optimization
> >
> > Thank you for this question. We clarify that the optimization tasks are not based solely on QM9-derived scaffolds. Instead, we use a combined set of scaffolds from both QM9 and ZINC250K, allowing us to incorporate a broader range of chemical structures, including both small and larger, more drug-like scaffolds. Specifically, 37 scaffolds are from QM9, and 63 scaffolds are from ZINC250K datasets. To improve clarity and transparency, we also added Appendix A.5, where we present the full set of 100 scaffolds used across the 20 optimization tasks, and share their sources. This includes their molecular weight distributions, oracle scores, and scaffold visualizations. Their distribution on molecular weight and their visualizations show that there are scaffolds with varying sizes, introducing different challenges in optimization.
> >
> > This mixed-source design enables us to evaluate the generalizability of CLaSMO across diverse structural contexts. As we detail in the newly added Appendix A.8, scaffold size plays a key role in optimization difficulty—especially under high similarity constraints. Smaller scaffolds tend to amplify the effect of each modification, presenting a greater challenge for preserving similarity, while larger scaffolds allow for more flexible edits.
> >
> > ## Q3 – Conditioning on Attachment Points Only
> >
> > Thank you for this comment. While our method relies on features of individual atoms, these features inherently capture important aspects of the local chemical environment. For instance, atom degree reflects the number of directly bonded neighbors, valence describes bonding capacity shaped by those neighbors, and ring membership indicates structural constraints around the atom. These localized yet context-aware descriptors help guide the generation of compatible substructures. We acknowledge that more complex tasks—such as structure-based drug design—may require incorporating broader scaffold-level context, which could be explored in future extensions. However, our experiments demonstrate that the current feature set is sufficient for the optimization tasks considered in this work.
> >
> > ## Q4 - Comparison to Diffusion-Based Methods
> >
> > Thank you for providing this valuable and relevant study. We have included it in our related works section. We understand that this methodology is designed for both de novo generation and molecular editing, offering considerable flexibility. We anticipate that, with its strong emphasis on sample efficiency, CLaSMO may perform better in low-budget optimization scenarios. On the other hand, the GenMol model stands out for its versatility and strong optimization performance in high-budget settings.

---

### Review · Reviewer_WWB2 · 2025-06-22

**Summary Of Contributions:**

This paper presents CLaSMO, a method that extends existing latent space molecular design approaches to scaffold decoration tasks. The key technical contribution is a sample-efficient Conditional Variational Autoencoder (CVAE) trained to generate small, synthetically-plausible substructures. Crucially, the generation is conditioned on the local atomic environment of a specific attachment point on the scaffold of interest and is then integrated with Latent Space Bayesian Optimization (LSBO) to efficiently search for optimal modifications. The LSBO performs a joint optimization over the substructure's continuous latent space and the discrete set of possible attachment points, aiming to improve a target property while maintaining structural similarity to the original molecule.

**Audience:**

Yes

**Broader Impact Concerns:**

Potential ethical implications are sufficiently addressed in the broader impact statement.

**Claims And Evidence:**

Yes

**Requested Changes:**

- The CVAE is trained on fragments from the QM9 dataset, which consists of small, simple molecules. This choice might bias the generator towards producing smaller, less complex substructures. It would be great to add a brief discussion in the Limitations section about how this data choice might affect the model's ability to discover optimal molecules when larger or more complex functional groups are required.
- What threshold constitutes a high similarity constraint depends on the context and featurization - could you explain the rationale behind choosing a Dice similarity of 0.5 as the high constraint threshold in this setting?
- The mixed kernel in Sec. 4.3.1 seems to use a single lengthscale. Could you clarify if separate lengthscales were used for the continuous and discrete components (and how they were chosen) or briefly justifying the current design choice?
- Eq. 2 appears to be missing parenthesis or a composition operator.
- The symbol $\mathcal{L}$ is used for both the CVAE loss (Eq. 1) and the set of labeled data (Sec. 3.2). I think it would improve clarity to use distinct notation for these two concepts.

**Strengths And Weaknesses:**

Strengths:
-  The core idea of operating in a conditional latent space determined by the local atomic environment of the attachment point is an interesting addition to existing molecular optimization methodologies, addressing the challenge of ensuring that a proposed modification is chemically sound and likely to bond successfully.
- The experimental design emphasizes sample efficiency by focusing on a low-budget (100 oracle calls) regime. This setup accurately reflects the constraints of real-world drug discovery and demonstrates the method's performance in resource-limited settings.
- The use of 20 different optimization oracles and three different similarity constraint levels constitutes a robust assessment of the method's performance in different application domains.
- The paper is well-written and easy to follow. The problem setup is well-defined and the methodology is explained clearly and logically.

Weaknesses:
- The claim of improving "real-world applicability" is primarily supported by in-silico Synthetic Accessibility (SA) scores. While CLaSMO seems to strike a balance between property optimization and synthesizability, it would be great to provide some more rationale or additional metrics in support of this claim.
- The CVAE is trained exclusively on fragments derived from the QM9 dataset, which consists of very small and simple molecules. While this aligns with the goal of small modifications, it may limit the model's ability to discover larger or more complex functional group (e.g. those with chlorine atoms). Is there a reason why the CVAE was not trained on more relevant BRICS fragments from e.g. ZINC or ChEMBL?
- The optimization is performed over a joint space of continuous latent vectors and discrete attachment points. The paper uses a simple kernel that treats each attachment point as independent (via a Kronecker delta). This approach may not scale well for large scaffolds with many potential attachment points, as it fails to leverage chemical similarity between different attachment points (e.g., two sp2 carbons in similar ring systems). Similarly, the currently used attachment point features are inherently local and might fail to capture broader steric and electronic information.

---

> ### Author Response · Authors · 2025-07-29
> **Updates on Synthesizability & Dataset Selection & Similarity Constraints (1/2)**
>
> We sincerely thank the reviewer for their constructive feedback, as well as for recognizing the novelty and clarity of our proposed CLaSMO framework. We also acknowledge the reviewer’s comments regarding potential limitations in our current setup, particularly concerning synthesizability, similarity constraints, and the modeling of attachment points. Below, we address each of the raised concerns and describe the corresponding additions and clarifications made to the manuscript.
>
> ## Weakness 1 - Limited Support for Real-World Synthesizability Claims
>
> Thank you very much for your comment. In the original manuscript, we primarily emphasized structural similarity to known molecules and supported our claim of enhanced real-world applicability through Synthetic Accessibility (SA) scores. We initially believed this would sufficiently support our claim. However, we fully agree that, given the centrality of this claim, it warrants further validation through complementary metrics.
>
> To address this, we have now incorporated Retrosynthetic Accessibility (RA) scores into our evaluation and updated the discussion accordingly in Section 5.2.3. The RA score provides a machine learning-based estimate of a compound’s synthesizability from the perspective of retrosynthetic planning. The results align well with the SA score analysis: both CLaSMO and MARS achieve favorable synthesizability across similarity constraints, reinforcing the claim that CLaSMO generates molecules that are both optimized and likely to be synthesizable in practice.
>
> ## Weakness 2 - Restricted Fragment Diversity Due to QM9-Only Training
>
> Thank you for your comment. Our primary motivation for using QM9 stems from its ability to yield compact, simple, and chemically interpretable BRICS fragments. These fragments are typically less than 100 Da in molecular weight, making them highly suitable for our modeling objective: introducing small, context-aware modifications to a given scaffold while maintaining high structural similarity.
>
> In contrast, BRICS decomposition of the ZINC250K dataset produces a significantly broader and heavier distribution of fragments. In the updated manuscript, we added a new discussion on this comparison to Section 5.1.1 and provided the molecular weight comparison of the fragments obtained from QM9 and ZINC250K datasets in Figure 4, where many ZINC-derived fragments exceed 100 Da and are structurally more complex. Using such fragments without filtering and training the generative model with such fragments leads to larger modifications to the input molecule during optimization, which often violates similarity constraints and hampers bonding compatibility.
>
> Nonetheless, we acknowledge the value of expanding the fragment vocabulary and have explored this in Appendix A.2 of the updated manuscript. Specifically, in Appendix A.2.1, we evaluate a model trained exclusively on unfiltered ZINC250K fragments. This variant performs poorly under high similarity constraints, as it frequently generates overly large substructures that significantly alter the scaffold.
>
> In Appendix A.2.2, we experiment with a hybrid dataset, combining QM9 fragments with filtered, size-compatible fragments from ZINC250K. This model preserves the compactness of the original setting while modestly increasing fragment diversity. Notably, it achieves comparable performance to the original QM9-only model in QED, troglitazone\_rediscovery, and mestranol\_similarity optimization tasks, despite no additional hyperparameter tuning. This indicates potential for its scalability and further improvement with proper model calibration and more extensive addition of such new fragments.
>
> ## Weakness 3 - Oversimplified Modeling of Attachment Points in Optimization
>
> Thank you for your fair criticism. We fully agree that treating attachment points as independent through a Kronecker delta-based categorical kernel imposes certain limitations. This simplification was adopted to ensure tractability and interpretability of the optimization in the current setting, where modeling attachment points as independent categories has proven sufficient for guiding sample-efficient optimization. However, we acknowledge that this part can be improved in future studies.
>
> Although our method focuses on single-atom features, these inherently reflect local context. For example, degree represents the number of neighboring atoms directly bonded to the target atom, valence captures how those neighbors shape the atom’s bonding capacity, and ring membership indicates whether the atom lies within a locally constrained ring structure. By highlighting these localized yet context-aware features, our approach effectively guides substructure generation. We acknowledge that more complex tasks would benefit from global structural features, which are currently absent but could be integrated in future work. Nonetheless, our experiments show that the present features suffice for the optimization tasks in this study.

---

> > ### Author Response · Authors · 2025-07-29
> > **Updates on Synthesizability & Dataset Selection & Similarity Constraints (2/2)**
> >
> > ## Requested Change 1 - Potential Bias from Training Only on Small QM9 Fragments
> >
> > Thank you for this question. As discussed in our response to Weakness 2, in the updated manuscript, we discuss our dataset selection and its implications in detail in Appendix A.2, where we also present preliminary experiments on property optimization using different dataset configurations. In the revised main text, we refer readers to this analysis in Section 5.1.1, and we additionally address this point in the Limitations section.
> >
> >
> > ## Requested Change 2 - Unclear Justification for High Similarity Threshold Selection
> >
> > Thank you for this question. In our setting, we define a high similarity constraint as a Dice similarity threshold of 0.5 based on an analysis of how scaffold similarity evolves during sequential fragment additions. After the reviewers' question, we included a discussion on this topic in Appendix A.8. As shown in Figure 13 in this section, Dice similarity can drop below 0.5 after just a few modifications—especially for smaller scaffolds, where each addition causes a relatively larger structural change. In contrast, larger scaffolds tolerate multiple modifications before falling below this threshold.
> >
> > Our input scaffolds include mix of small and large scaffolds (we also added the characteristics and visualizations of the input scaffolds to Appendix A.5). Given that our dataset includes both small and large scaffolds, we found that setting the threshold to 0.5 offers a good trade-off: it permits at least one meaningful modification for small scaffolds while still preserving a reasonable level of similarity for larger ones. Thus, the choice of 0.5 is both empirically grounded and size-aware, ensuring that the similarity constraint is not overly restrictive for smaller molecules, yet remains effective for controlling structural deviation across scaffold sizes.
> >
> > ## Requested Change 3 - Insufficient Explanation of Mixed Kernel Design in BO
> >
> > Thank you very much for pointing this out. We use separate lengthscales for the continuous and discrete components. In the updated manuscript, we have clarified this by introducing distinct notation for each lengthscale. These parameters are learned by maximizing the log marginal likelihood during Gaussian Process training. We have added the relevant explanation to Appendix~A.3.
> >
> >
> > ## Requested Change 4 - Missing Notational Clarity in Equation 2
> >
> > Thank you very much for pointing this out. We have updated Equation 2 to include the missing parentheses.
> >
> > ## Requested Change 5 - Ambiguous Use of Symbols Across Different Contexts
> >
> > Thank you very much for pointing this out. To avoid confusion, we have updated the notation for the loss function to J, ensuring it is distinct from the symbol used for the set of labeled data.

---

> > > ### Comment · Reviewer_WWB2 · 2025-08-07
> > > **Acknowledgement of the author response**
> > >
> > > I would like to thank the authors for the detailed response. The provided clarifications and additions to the manuscript address all of my questions and concerns.

---

### Review · Reviewer_vekD · 2025-07-25

**Summary Of Contributions:**

The paper introduces a new approach to tackle molecular scaffold optimization. The authors use a combination of a conditioned VAE and BO in its latent space to find new, additional substructures for specific attachment points. To this end, the authors construct a new dataset of suitable molecular substructures from existing molecules using the BRICS fragmentation method. This ensures that the VAE will see synthetizable useful substructures during training. During inference time, this work uses BIO in the latent space of the VAE to produce suitable substructures to attach to the scaffold.

**Audience:**

Yes

**Broader Impact Concerns:**

Broader Impact statement is present.

**Claims And Evidence:**

Yes

**Requested Changes:**

Questions:
- What fraction of CVAE generations cannot bond or are chemically invalid?
- What acquisition function is used for the GP? How is it optimized (discrete and continuous components)?


Requested changes:
Adressing the questions and eliminating the weaknesses 2-5

**Strengths And Weaknesses:**

Strengths:
- The proposed approach seems very suitable to tackle the problem. I like creating the dataset for the VAE
- The empirical evaluation shows that the method works very well on most tasks
- Making the project available for chemists in a simple way is great


Weaknesses:
- The novelty is not super high. To me, it seems like the contribution lies mainly in the dataset creation since the CVAE and the LSBO are not super novel.
- Using only QM9 derived substructures might be limiting diversity etc. I would like to see how the approach scales when using a larger variety of data
- Some details are missing regarding the GP: see questions.
- Ablation study: while I like that the conditioning is ablated, I would like to also see the effect of e.g. the LSBO in total (so vs random search), the Kernel selection, potentially a different fragmentation scheme compared to BRICS etc.
- Since MARS (a rather old technique) has a strong performance I would like to see an even more detailed comparison: a longer paragraph on the approach in related work, a section (maybe in the appendix) to compare where each method excells and where the other fails so that the user knows what approach to pick etc.

---

> ### Author Response · Authors · 2025-07-29
> **Updates on Dataset Selection & Ablation Study and Bonding Success (1/2)**
>
> We thank the reviewer for their constructive criticism and thoughtful assessment of our work. We appreciate the recognition of the value in our dataset construction and the practical applicability of our method for chemists. Below, we address each of the reviewer’s concerns and suggestions in detail.
>
> ## Weakness 1 - Concerns on Novelty
>
> Thank you for your comment. We agree that the usage of CVAE and LSBO is not new in the context of molecular design. However, proposed methodologies in the literature that use LSBO in the latent space of VAE-based models are designed to make from-scratch generations, where a totally new molecule is generated instead of editing existing molecules.  They are not designed for editing the given molecule. In this sense, our proposal positions itself as the novel LSBO-based approach for molecular editing.
>
> ## Weakness 2 - Concerns on QM9 Dataset Selection
>
> Thank you for your comment. In the updated manuscript, we included a detailed discussion on our dataset selection process in Section 5.1.1. We demonstrate the BRICS fragments found from the QM9 and ZINC250K datasets, and compare their molecular weights. As the ZINC250K dataset includes fragments that are heavy and large, and the QM9 dataset provides small fragments that are mostly less than a molecular weight of 100 Dalton, we selected the QM9 dataset for the sake of completeness and simplicity.
>
> However, one can, of course, filter smaller fragments from different publicly available datasets and use them to train our proposed method to introduce a larger variety. We dedicated a new section in the Appendix, Appendix A.2, for this alternative. In Section Appendix A. 2.1, we evaluate the case where only ZINC250K fragments are used without any filtering. We show that this case fails especially when there are similarity constraints, as this model tends to generate larger substructures that introduce larger modifications to the input molecules. In Section Appendix A. 2.2, we evaluate another alternative where smaller fragments from the ZINC250K dataset are used in addition to the original QM9 fragments. In this setting, we observe that it performs closely to the original setting in QED, troglitazone\_rediscovery, and mestranol\_similarity optimization tasks, and outperforms other baselines, showcasing the proposed methods’ potential to be scaled. As these results are obtained without fine-tuning of the hyperparameters of this new model, we can expect it to perform better with careful tuning of its CVAE and LSBO-based hyperparameters.
>
> ## Weakness 4 - Additional Ablation Studies
>
> Thank you for this suggestion. In the updated manuscript, we have added an ablation study comparing LSBO to Random Search (RS) under the same generation budget, and we revised the ablation study section to be consistent with the other evaluation metrics considered in this study. We now compare LSBO+CVAE (original CLaSMO setting), RS+CVAE, and LSBO+VAE in mestranol\_similarity, QED, and troglitazone\_rediscovery oracle optimization problems. This comparison, as shown in Section 5.3, clearly demonstrates that the original setting outperforms these alternatives, where the impact of using the conditioning mechanism stands out.
>
> We agree that further ablations—such as exploring alternative kernel designs or fragmentation strategies—could provide additional insights. However, these directions require a more thorough investigation. Our current goal was to validate the key design components—namely, the combination of LSBO with conditional latent space modeling—which we believe is well supported by the results presented. We appreciate the reviewer’s suggestions and consider them promising directions for future work.

---

> > ### Author Response · Authors · 2025-07-29
> > **Updates on Dataset Selection & Ablation Study and Bonding Success (2/2)**
> >
> > ## Weakness 5 - More Discussion on MARS
> >
> > Thank you for this comment. Following the reviewers’ suggestions, we expanded the discussion on MARS in the related works section. We also added a new section to Appendix A.10, where we discuss that MARS can be a good choice for multi-property optimization under relaxed similarity constraints, while CLaSMO can be preferred in other cases.
> >
> > ## Requested Change 1 - What fraction of CVAE generations cannot bond or are chemically invalid?
> >
> > Thank you for this question. As we think that the answer to this question should be given in the manuscript, we added a new section to Appendix A. 9 titled “Analysis on Bonding Success During LSBO”, where we demonstrate the bonding success in each 20 oracle optimization tasks.
> >
> > In our bonding success analysis for 20 oracle optimization tasks, we observed that the CVAE model in CLaSMO produces chemically valid and bondable substructures with extremely high reliability. Within the first 10 iterations of each optimization run, more than 99\% of the generated substructures successfully bond with the scaffold, and this success rate remains above 98\% within the first 25 iterations. Even up to iteration 50, the bonding success rate exceeds 80\% across all tasks. Thus, the fraction of CVAE generations that cannot bond or are chemically invalid is less than 1–2\% in the early stages of optimization and remains relatively low throughout. Failures become more frequent in later iterations due to increased exploration of uncertain latent regions and reduced scaffold bonding capacity (as optimization goes to later stages and the input scaffold gets updated, number of atoms within the molecule with additional capacity to form a bond decreases, if not exhausts), but overall, the bonding reliability of the model is consistently high.
> >
> > ## Requested Change 2 - What acquisition function is used for the GP? How is it optimized (discrete and continuous components)?
> >
> > Thank you for the question. We use the Upper Confidence Bound as the acquisition function, as described in the last paragraph of Section 4.3.
> >
> > The search space consists of both continuous variables from the latent space of the CVAE and an integer variable representing the index of discrete atom types. To optimize the acquisition function, we perform gradient-based optimization over a relaxed continuous domain, where the integer component is temporarily treated as a continuous variable. This relaxation enables the use of standard multi-start gradient-based optimization techniques over the full (mixed) input space.
> >
> > After the optimization process identifies a continuous solution, the value corresponding to the categorical dimension is rounded or floored to the nearest valid index. This post-processing step maps the relaxed continuous solution back to the discrete domain, enabling proper interpretation of the categorical choice while maintaining the efficiency of continuous optimization. This method avoids the computational overhead of explicitly enumerating categorical variables and integrates seamlessly into acquisition optimization routines that support automatic differentiation.
> >
> > The idea of handling mixed integer and continuous variables via rounding has been discussed in the literature. One such study is:
> >
> > Eduardo C. Garrido-Merchán, Daniel Hernández-Lobato,"Dealing with categorical and integer-valued variables in Bayesian Optimization with Gaussian processes," Neurocomputing
> >
> > We have added this explanation of our optimization strategy for the mixed search space to Appendix~A.3 for completeness.

---

### Decision · Action_Editor_qn4i · 2025-08-30

**Recommendation:** Accept as is

**Audience:**

Yes

**Audience Explanation:**

There is a clear need for sample-efficient methods for de novo drug design due to the sometimes high computational complexity of reward scores and the obviously high cost of wet experimentation.

**Claims And Evidence:**

Yes

**Claims Explanation:**

The paper proposes a novel method for de novo molecular generation that combines CVAE and (latent) Bayesian optimization. The core focus is on generating structures in a sample-efficient manner, which I would view as inspired by work on scaffold hopping and genetic algorithms for de novo drug design.

The main claim is that the method is more sample efficient than alternative methods, in particular due to narrowing the search to more synthetically accessible compounds.

The work has made improvements to the evaluation since the last submitted version that now adequately support the thesis. I would like to thank the Authors for their work. The method generally outperforms other methods in low data regime, including methods that also take into account fragmentation. The ablation support the overall motivation behind the work. The results are further supported showing that the fragmentation contributes positively to both heuristic and retrosynthesis based complexity scores.